# Tunable encapsulation of sessile droplets with solid and liquid shells

Rutvik Lathia [1,4], Satchit Nagpal[1,4], Chandantaru Dey Modak [1], Satyarthi Mishra[1], Deepak Sharma[1], Bheema Sankar Reddy[1], Pavan Nukala [1], Ramray Bhat [2,3] & Prosenjit Sen [1,3] ✉

Droplet encapsulations using liquid or solid shells are of significant interest in microreactors, drug delivery, crystallization, and cell growth applications. Despite progress in droplet-related technologies, tuning micron-scale shell thickness over a large range of droplet sizes is still a major challenge. In this work, we report capillary force assisted cloaking using hydrophobic colloidal particles and liquid-infused surfaces. The technique produces uniform solid and liquid shell encapsulations over a broad range (5–200 μm shell thickness for droplet volume spanning over four orders of magnitude). Tunable liquid encapsulation is shown to reduce the evaporation rate of droplets by up to 200 times with a wide tunability in lifetime (1.5 h to 12 days). Further, we propose using the technique for single crystals and cell/spheroid culture platforms. Stimuli-responsive solid shells show hermetic encapsulation with tunable strength and dissolution time. Moreover, scalability, and versatility of the technique is demonstrated for on-chip applications.

The use of droplets has rapidly progressed for a wide range of applications such as microreactors[1,2], biochemical assays[3], sensors[4,5], optofluidic resonators[6], and crystal growth[7,8]. However, such applications commonly suffer from problems such as contamination, substrate dependence, and rapid evaporation[9,10]. Higher evaporation and contamination of the droplets significantly affect the cell viability in bioreactors[11], reliability of chemical reactions[12], quality of crystal produced[13], and stability of droplet sensors[6]. Encapsulating the droplets inside an immiscible liquid[14,15] or solid medium[16,17] is a way to deal with these problems. Despite significant progress in droplet-based technologies, tunable microscale shell encapsulation of millimeter-scale sessile droplets remains primarily unaddressed.

Tunable encapsulation using immiscible liquid shells will enable regulation of the droplet's evaporation rate without any complex and special setup. Control of evaporation is of utmost importance in crystal growth, as uncontrolled evaporation results in amorphous or polycrystalline growth, precipitation, and defects[13,18]. Furthermore, the evaporation rate significantly affects the number, size, and density of the crystals produced[19,20]. In droplet-based cell cultures, evaporation

changes the osmolality of the media and severely affects cell viability[11]. Similarly, liquid encapsulation inside a solid shell is utilized in many vital areas, such as drug delivery, agriculture, food, and textiles[21,22]. Wide thickness tunability of such a solid encapsulation shell is required to tune its strength.

Several standard methods for droplet encapsulation include microfluidics, molding, injection, and dip coating[14,17,23,24]. The microfluidic techniques are complex, require microfabricated devices, and a long setup time. Additionally, they are limited to smaller droplet sizes (<300 μm) with a limited tunability in encapsulation thickness (1–50 μm)[25,26]. In contrast, other approaches suffer from non-uniform shells[14,27]. Further, they are limited to higher shell thickness (>0.5 mm) with limited control[14,17,23]. A technique of droplet encapsulation that: (i) covers a wide droplet size range (from hundreds of micrometers to several millimeters) and; (ii) provides a widely tunable shell material and thickness can address many long-standing problems of sessile droplet applications.

Capillary-driven cloaking of the sessile droplet can be a possible solution. Upon deposition of droplets on oil-infused surfaces, the oil

[1]Centre for Nano Science and Engineering, Indian Institute of Science, Bangalore 560012, India. [2]Department of Developmental Biology and Genetics, Indian Institute of Science, Bangalore 560012, India. [3]Department of BioSystems Science and Engineering, Indian Institute of Science, Bangalore 560012, India. [4]These authors contributed equally: Rutvik Lathia, Satchit Nagpal. ✉e-mail: prosenjits@iisc.ac.in

from the surface tends to rise and cover the droplets[28]. Oil encapsulates aqueous droplets if the spreading factor of oil on water ($S_{ow}$) is positive ($S_{ow} = \gamma_w - \gamma_o - \gamma_{ow} > 0$), where, $\gamma_w$, $\gamma_o$, and $\gamma_{ow}$ are the surface tension between water-air, oil-air, and oil-water interfaces, respectively. Oil-based cloaking, conceptually, can reduce the evaporation rate and protect against contamination. However, in practice, the reduction in evaporation rate is minimal as the thickness of the oil layer at the top of the droplet is in the hundreds of nanometer regime[27]. Since the thickness of the oil cloaking depends on the interplay between various physical forces such as surface energies, gravity, and long-range forces, it is theoretically impossible to have a uniform and tunable cloaking layer with the oil layer alone.

We report a method of tunable cloaking of droplets with the help of nanometer to micrometer scale particles. The droplets were first coated with particles (commonly known as liquid marbles), and then the stable cloaking layer was formed by exposing them to oil-infused surfaces. The particle coating promotes and stabilizes the immiscible liquid film over the droplet. Encapsulation with both solid and liquid phases has been demonstrated. Encapsulation thickness depends on the particle size, and tunability in the range of 5–200 μm has been achieved for the size of sessile droplets ranging from 14 nL–200 μL. For liquid encapsulation, we show up to a 200x increase in the lifetime of the droplet against evaporation. Lifetime tunability ranges from 1.5 h to 12 days for a 10 μL droplet encapsulated with various particle loading and oils. Additionally, we have shown application in single crystal growth of copper sulfate, Rochelle salt, sodium nitrate, and lysozyme protein. The feasibility of our microreactors for biological applications was tested by growing human and yeast cells in a hanging droplet configuration. Solid capsules respond to external stimuli (temperature changes), and their behavior can be tuned by varying encapsulation thickness. Further, using the stimuli-responsive oils makes on-demand uncoating and merging possible.

## Results and Discussion
### Composite droplet structure
Coating particles on oil-covered droplets have been used to demonstrate composite liquid marble structures[29]. Tunability is not possible as the oil layer is coated first. Oil layers on bare droplets have a non-uniform thickness (Supplementary Fig. 1). This results in the formation of non-uniform clumps during the subsequent particle coating (Supplementary Fig. 2). Such clumps harm the integrity and effectiveness of the composite LM (Supplementary Fig. 3). Similarly, other methods of oil coating, such as concentric needles-based methods, also suffer from non-uniformity in shell thickness due to density and viscosity mismatch[30]. Further, the same composite LM work demonstrated that coating particles before oil exposure is not trivial as it displaces the particles and ruptures the particle coating (See Supplementary Movie 1)[29]. When a liquid marble (LM) is exposed to oil from the top or bottom, the capillary forces drive the oil to encapsulate the liquid marble. The hydrodynamic drag force on the particle scale as $\eta_o R_p v$, where $\eta_o$ is the oil viscosity, $R_p$ is particle radius, and $v$ is the capillary velocity of oil. The oil encapsulation velocity $v$ is observed to be ≈ 10 mm s⁻¹. Thus, hydrodynamic drag on 35 μm particles and a 10 cSt oil is in the order of ≈10⁻⁸ N. The adhesive force on a particle scales as $\pi \gamma_{ow} R_p \cos^2(\theta_{ow}/2) \approx 10^{-8}$ N, where $\theta_{ow}$ is the contact angle of the oil-water interface with the particle (here, $\gamma_{ow}$ = 42 mN m⁻¹, $R_p$ = 17.5 μm, and $\theta_{ow}$ = 170°)[31]. Due to similar magnitudes, the hydrodynamic forces on the particles can overcome the surface forces holding the particles on the droplet interface. Detailed comparisons with the composite LM have been listed in Supplementary Notes 1, 2 and Supplementary Table 1.

To have a tunable and uniform layer over a sessile droplet, Liquid marble is first prepared by rolling a liquid droplet on a PTFE powder bed[32]. Prepared liquid marbles are then gently placed on the oil-infused nanostructured surface. Depending upon the oil type and property, the oil from infused surfaces rises and covers the whole liquid marble.

The nanostructured surfaces offer a considerable drag to the oil flow, which ensures lower cloaking velocity. The velocity of oil rise is significantly slower (≈10⁻² mm s⁻¹). The hydrodynamic drag on the particles is reduced to ≈10⁻¹¹ N. This is around three orders of magnitude lower than the particle adhesion at the interface. Thus, enabling a stable LM cloaking without any disruption of the particle coating.

Figure 1a and Supplementary Movie 2 represent the cloaking dynamics of oil over LM, where the oil can be seen to encapsulate the droplet as time progresses. Due to the effective surface tension of LMOI, the liquid marble achieves a finite contact angle with the surface after some time. Fig. 1b shows the schematic of oil-coated liquid marble. Oil also encapsulates a bare droplet, but the thickness of the coated oil layer is non-uniform and changes with the droplet height and is known to be in the nanometer regime at the top[28,33]. However, in the case of LM, the oil-coating is uniform and is governed by particle size. Figure 1c represents the fluorescence image of the oil coating on the plane 1 mm above the substrate. Oil thickness is ≈ 50 μm for LMOI, compared to 6 μm for the bare droplet. For LMOI, oil thickness is in the range of PTFE particles of size (35 μm). Moreover, as shown in Fig. 1e, this technique is suitable for a wide range of droplet sizes (300 μm to 7.5 mm diameter).

### Condition for LMOI stability
As the LM is encapsulated in oil, it spreads due to the changes in the effective surface energies. The LMOI reconfigures to achieve a different contact angle than the LM because of the effective surface energy change, as seen in Fig. 1a. The value of the LMOI contact angle on the surface also affects the stability of LMOI significantly. The static contact angle for LMOI can be defined as $\cos\theta_{LMOI} = (\gamma_o - \gamma_{ow})/\gamma_{LMOI}$. If the value of $\theta_{LMOI} > \theta_{cr}$, then the successful formation of LMOI is possible. However, for $\theta_{LMOI} < \theta_{cr}$, the LMOI spreads on the infused surface and generates cracks in a particle coating. Here, $\theta_{cr}$ is defined as the critical LMOI contact angle that determines whether the LMOI will remain stable and crack-free on the surface or not. Our experiments indicate $\theta_{cr} \approx 90°$. Cracks happen because of the larger surface area of the LMOI interface at the lower $\theta_{LMOI}$, which generates higher inter-particle distance and, thus, cracks (Supplementary Note 3). Such cracks are undesired as they create non-uniform coating across the interface. This implies that the value of $\Gamma = \gamma_{ow} - \gamma_o$ should be positive (Fig. 1d). The interfacial tension of various oils is represented in Supplementary Table 2. A combination of oil and particle properties that satisfies all mentioned conditions would result in a stable and thick oil coating over the liquid marble.

### Control of encapsulation thickness
The thickness of the cloaking oil layer depends on the particle size. To study this, we consider the case of solid encapsulation. The cloaked layer can be converted into a solid capsule by using a phase change material (e.g., wax) instead of oil. In this case, the cloaking is achieved at a higher temperature. When the temperature is reduced, a solid capsule is obtained. Fig. 1f, g represents the solid capsules made of wax with PTFE and hydrophobic glass beads, respectively. The encapsulation thickness is observed to be proportional to the size of the particles, as seen in Fig. 1h. The uniformity of the shell is evident in Fig. 1i, where the thickness of the shell at three different regions (bottom left, top, and bottom right) was measured as represented in Supplementary Fig. 4. The mean thickness deviation within a droplet measured over three different points and three independent measurements is 7.4%. Similarly, the thickness of the shell can be tuned by varying the particle loading (mass loading) across the LM surface (Supplementary Fig. 5). For smaller particle sizes (<1 μm), the shell thickness is larger due to the agglomeration of the particles (Supplementary Fig. 7). Such agglomeration in 800 nm particle size results in a 6 μm thick shell. Additionally, the value of the particle contact angle at the oil-water ($\theta_{ow}$) interface has to be higher (>150°) to ensure the shell's thickness

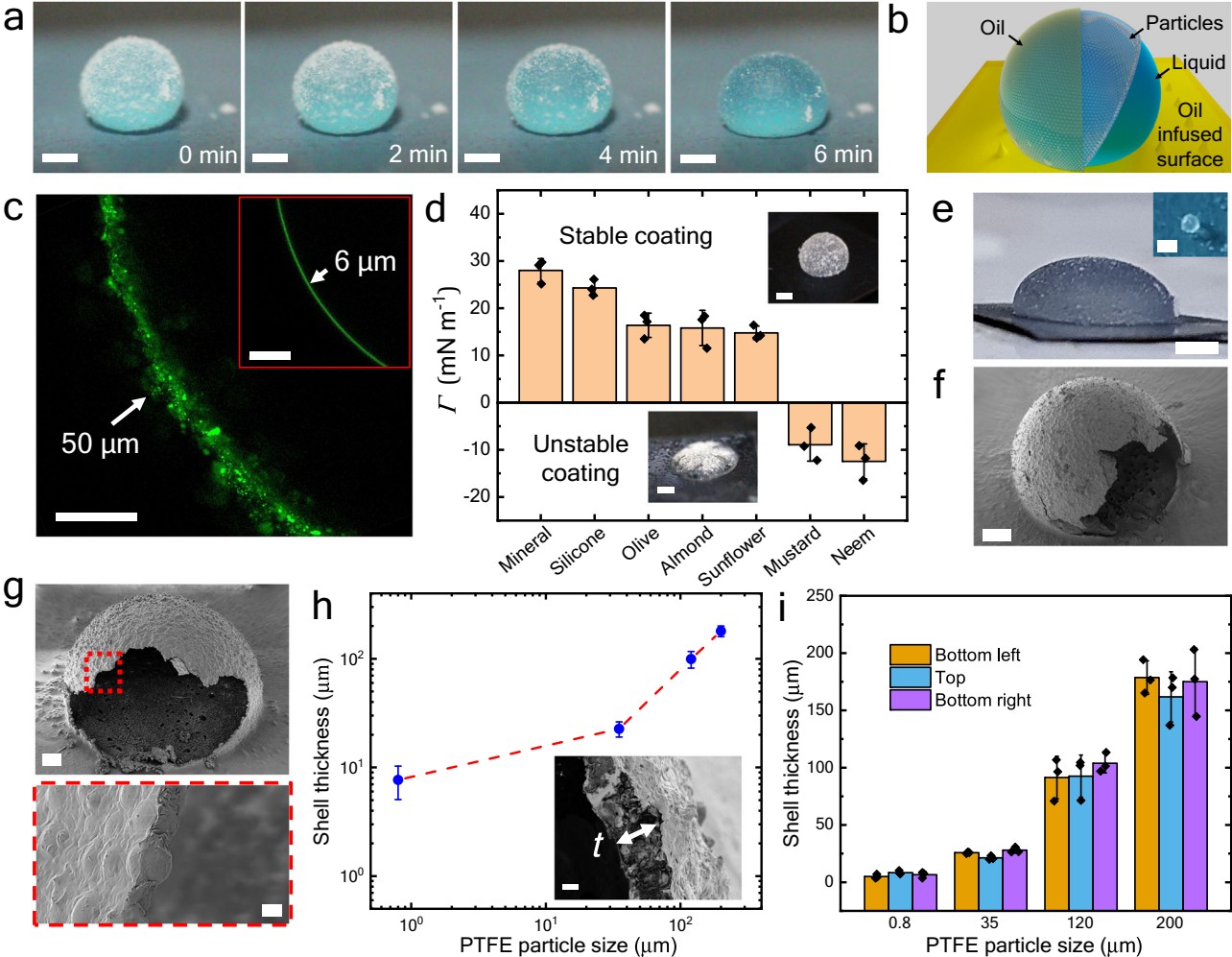

**Fig. 1 | The tunable liquid and solid shells using the LMOI method. a** Liquid Marble on silicone oil (50 cP viscosity) infused surface in which oil rises to cover the LM as time progresses. The volume of the inner liquid is 10 μL, and the particle size is 35 μm PTFE. Entire LM cloaking can be seen in the initial 2 to 4 min. Scale bar = 1 mm. **b** Schematic representation of LMOI (Liquid marble on an oil-infused surface) where the coating of particle and oil is represented on an OI (liquid/oil-infused) surface. **c** A confocal microscope image shows the thickness of the silicone oil coating for LM and droplet (inset). Scale bars = 200 μm. **d** Stability of LMOI for different oils. A negative value of $\Gamma = \gamma_{ow} - \gamma_{o}$ represents an unstable coating where crack formation in LMOI is present (inset: LMOI – neem oil). A positive value of $\Gamma = \gamma_{ow} - \gamma_{o}$ represents a stable coating where no crack formation in LMOI is observed (Inset: LMOI – mineral oil). Scale bar = 1 mm. Measurements were carried out on $n = 3$ independent samples, and data are presented as mean values ± SD. Black dots represent individual data points. **e** LMOI prepared from 200 μL liquid volume. Scale bar = 2 mm. Inset: LMOI prepared from 14 nL

liquid volume. Scale bar = 300 μm. **f** SEM image of a half-cut dried capsule based on PTFE particles. Scale bar = 400 μm. **g** SEM image of a half-cut dried capsule based on hydrophobic glass particles (35 μm). Scale bar = 400 μm. The red-dotted region represents the shell wall where spherical glass beds are visible through the cloaked layer of wax. Scale bar = 20 μm. **h** Shell thickness is represented as a function of particle size. Inset: SEM image of the PTFE shell wall. Scale bar = 40 μm. Measurements were carried out on $n = 3$ independent samples, and data are presented as mean values ± SD. **i**, The uniformity of the shell wall for different PTFE particle sizes. Measurements were carried out at 3 different regions, namely, the bottom left (orange color), top (blue color), and bottom right (purple color) parts of the capsule on $n = 3$ independent samples, and data are presented as mean values ± SD. Black dots represent individual data points. A detailed depiction of the locations is given in Supplementary Fig. 4. Source data are provided as a Source Data file.

to be approximately equal to the particle size. In PTFE's case, the $\theta_{ow} = 170°$ and 165° in silicone oil and mineral oil case, respectively (Supplementary Fig. 8).

## Nanostructured surface coating

Another important parameter is the hydrophobicity of the oil-infused surface. The nanostructures for the oil-infused surface were fabricated by a well-known method[34]. The nanostructured surface is then hydrophobized with stearic acid, and oil is infused inside it. The degree of hydrophobization is important. Higher surface tension oil, such as mineral oil, cannot be infused properly if the surface energy of the substrate is very low (i.e., coated with Teflon having $\theta_{SHP}$>165°). The $\theta_{SHP}$ is the contact angle of the water droplet over the prepared

superhydrophobic (SHP) surface. A stearic acid-coating ($\theta_{SHP}$≈150°) provides a better surface for the infusion of higher surface tension oils (> 30 mN m$^{-1}$).

## Dynamics of cloaking

The encapsulation thickness is observed to be approximately equal to the particle (or agglomerate) thickness. Hence, we assume that most of the flow is primarily through the particles while very little flow is there over the particles. The tracking of contact line in optical microscope reveals the major flow in between the particles and not above them (Supplementary Figs. 9, 10 and Supplementary Movie 3). The SEM of the shell also imparts a similar conclusion, where some part of the particle can be seen through the shell layer (Supplementary Fig. 11).

Based on the above observations, we have modelled the cloaking as a single-step process with oil thickness similar to the particle (agglomerate) thickness, as shown in Supplementary Fig. 12.

The detailed solution of the dynamics is given in the Supplementary Note 4. The driving force for oil rise over the liquid marble is capillary force ($F_c$). Drag offered by the nanostructures in the oil-infused surface ($F_d$) and weight ($W$) of the oil are the resistive forces acting against the oil rise (Supplementary Fig. 12a). At a small scale, capillary and drag forces are much stronger than the weight of the oil[35]. Thus, the balance of the forces can be given as

$$F_c = F_d \qquad (1)$$

The effective surface energy of the LM and LMOI can be derived by considering individual contributions from water, particle, oil, and interactions between them. Considering $n$ number of spherical solid particles with radius $R_p$ covering the droplet with radius $R$. The schematic representation of process dynamics is given in Supplementary Fig. 12b. As represented in Supplementary Fig. 12b, the whole system can be divided into three major parts, the base, LMOI (up to which oil rise happened; up to height $y$), and LM (where still oil covering has not happened; height ($h$-$y$), where $h$ is the initial height of the LM). During the oil rise, the LMOI settles, and the contact angle with the surface changes; thus, the dynamic contact angle, $\theta_d$ is defined. The effective interfacial energies ($E$) of these individual parts can be defined as

$$E_{LM}(y) = S_{LM}\gamma_w\left(1 - \frac{n\pi R_p^2 \theta_w}{S_{tot}}\right) + 2\pi R_p^2 n_{LM}(\gamma_p(1 - \cos\theta_w)$$
$$+ \gamma_{pw}(1 + \cos\theta_w)) \qquad (2)$$

$$E_{base}(y) = S_{base}\gamma_{ow}\left(1 - \frac{n\pi R_p^2 \theta_{ow}}{S_{tot}}\right) + 2\pi R_p^2 n_{base}(\gamma_{po}(1 - \cos\theta_{ow})$$
$$+ \gamma_{pw}(1 + \cos\theta_{ow})) \qquad (3)$$

$$E_{LMOI}(y) = S_{LMOI}\gamma_o + S_{LMOI}\gamma_{ow}\left(1 - \frac{n\pi R_p^2 \theta_{ow}}{S_{tot}}\right) + 2\pi R_p^2 n_{LMOI}(\gamma_{po}(1 - \cos\theta_{ow})$$
$$+ \gamma_{pw}(1 + \cos\theta_{ow})) \qquad (4)$$

Where the $E_{LM}$, $E_{LMOI}$, and $E_{base}$ are the interfacial energies of LM, LMOI, and base parts respectively. These energies were calculated by adding the interfacial energies of individual components (oil, particles, air). Surface areas of the individual parts are the base ($S_{base}$), LMOI ($S_{LMOI}$), LM ($S_{LM}$), and total surface area ($S_{tot}$). Here $n_{LM}$, $n_{LMOI}$, and $n_{base}$ are the total number of particles on the LM, LMOI, and base, respectively. Additionally, $\theta_w$, $\gamma_p$, $\gamma_{pw}$, $\theta_{ow}$, and $\gamma_{po}$ are the contact angle of the water interface with the particle, the surface energy of the particle-air interface, the surface energy of the particle-water interface, the contact angle of the oil-water interface with the particle, and the surface energy of the particle-oil interface, respectively (Supplementary Fig. 12c, d).

Thus, the total energy variation with $y$ is given by

$$E(y) = E_{base}(y) + E_{LMOI}(y) + E_{LM}(y) \qquad (5)$$

The total energy is observed to vary linearly. Hence, the energy difference between the initial and final state ($\Delta_{LMOI}$) drives the oil rise (Supplementary Fig. 13). The initial state is $E_i = E(0)$, and the final state is $E_f = E(h)$. Therefore, the driving capillary force can be given by

$$F_c = -\frac{dE}{dy} \approx \frac{\Delta_{LMOI}}{h} \approx \frac{E(0) - E(h)}{h} \qquad (6)$$

The viscous drag can be approximated using Poiseuille's law. The velocity gradient in the oil-infused surface is in the order of $\eta_o \, V \, p^{-1}$ where $V$ and $p$ are the velocity of oil, and the pitch between the nanostructures, respectively. The dissipation occurs over the area $\approx Dy$; thus, the viscous drag force is given by

$$F_d \approx \frac{\eta_o V D y}{p} \qquad (7)$$

Balancing the capillary force with the viscous drag force and putting the velocity of the oil rise as $V = y \, t^{-1}$ where $t$ is the time instance of rise, gives the rise height ($y$) as

$$y \approx \left(\frac{p\Delta_{LMOI}}{\eta_o Dh}t\right)^{0.5} \qquad (8)$$

Figure 2a, b represents the rising height of oil over an LM with time. The clear difference is present according to the viscosity of the oil utilized in making LMOI. The nondimensionalization of Eq. (8) reveals the time scale responsible for the oil rise in the LMOI system ($\tau_{LMOI}$). Thus, normalizing the rise height ($y$) by the LMOI height ($h$) gives Eq. (9)

$$\frac{y}{h} = c\left(\frac{t}{\tau_{LMOI}}\right)^{0.5}, \text{ where } \tau_{LMOI} = \frac{\eta_o Dh^3}{p\Delta_{LMOI}} \qquad (9)$$

Where $c$ is the constant of proportionality. As shown in Fig. 2c, the normalized data collapses into a single curve. Additionally, the slope of the data is approximately 0.5, which further validates Eq. (9). Fitting of Fig. 2c data with Eq. (9) reveals the value of $c = 0.323$.

Apart from the dynamic rise, the final and initial energy of the LMOI configuration also gives the necessary static condition for oil rise. For successful cloaking, the $\Delta_{LMOI} > 0$ conditions should be satisfied. For silicone oil case with 35 μm PTFE particles, the value of $\Delta_{LMOI} \approx 0.351$ μJ, confirming complete cloaking of the LM. For similar conditions, mineral oil has $\Delta_{LMOI} \approx 0.286$ μJ. In comparison, for bare droplet cloaking, the $\Delta_{ow} = 4\pi R^2 S_{ow}$ values for silicone and mineral oil are 0.242 and 0.007 μJ, respectively. Liquids with higher positive $\Delta_{LMOI}$ values spread (or rise) more easily on the LM surface compared to those with lower $\Delta_{LMOI}$ values. This should result in a stronger driving force for cloaking and a faster spreading (rise) over the LM surface. Due to its higher $\Delta_{LMOI}$ value, silicone oil has a greater affinity for cloaking the LM surface than mineral oil. Additionally, higher values of $\Delta_{LMOI}$ compared to $\Delta_{ow}$ for both silicone and mineral oil suggest a stronger affinity of oil to rise in the case of LM than the bare droplet.

## Resistance to evaporation

The microscale thick layer of oil on the droplet drastically reduces the droplet's evaporation rate. The temporal evolution of normalized droplet mass is represented in Fig. 3a, where liquid marble over an oil-infused surface (LMOI) shows an increased lifetime. We can control the evaporation rate through the thickness of the cloaking layer. It also depends upon the oil used. By using mineral oil, the lifetime of a 10 μL droplet can be increased up to 12 days, which is about 200 times more than a bare droplet. Silicone oil ($\eta_o = 5$ cP) results in a lifetime of approximately 26 h for a similar volume droplet (Fig. 3a). Even with relative humidity as high as 95%, the bare droplet evaporates in 10 h, while LMOI takes nearly 300 h even at low humidity (50%) (Supplementary Fig. 3). The reduction in evaporation rate can be attributed to three effects. First, the water vapor must pass through a layer of oil to evaporate. Second, the constriction of the diffusion path due to submerged particles. And third, the reduction in the effective surface area for evaporation because of the particle at the interface. However,

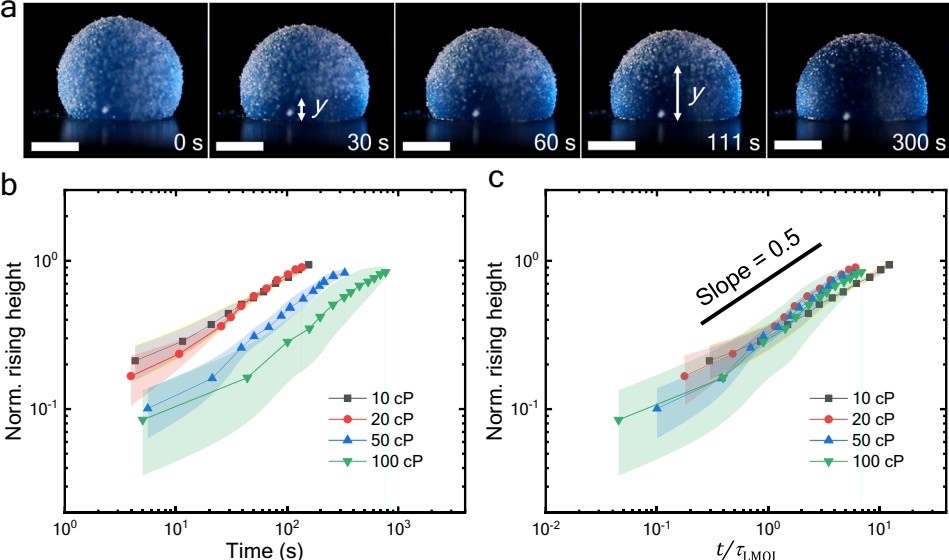

**Fig. 2 | The dynamics of oil-cloaking. a** The snapshots of oil rise in 10 μL LM with 35 μm PTFE particles and 50 cP silicone oil. Scale bar = 1 mm. **b** The evolution of normalized rising height (*y/h*) with time for different silicone oil viscosity (10 cP – black color, 20 cP – red color, 50 cP – blue color, and 100 cP – green color). Measurements were carried out on *n* = 3 independent samples, and data are presented

as mean values ± SD. **c** The scaled graph collapses into a single curve with a slope of 0.5 (black line) for all viscosity. Measurements were carried out on *n* = 3 independent samples, and data are presented as mean values ± SD. Source data are provided as a Source Data file.

as the particles are oleophilic, most of the particle is submerged in oil, while a tiny part touches the oil-water interface. Hence, only the first two effects play a major role in controlling the evaporation rate.

The position of the particle with respect to the oil-water interface is decided by the contact angle of the oil-water interface with the particle ($\theta_{ow}$). If $\theta_{ow}$>90°, a larger particle area is submerged in the oil, and if $\theta_{ow}$<90°, most of the particle area is submerged in the water. The value of $\theta_{ow}$ is found to be 170°, confirming less than 10% of the particle area is submerged in the water (Supplementary Fig. 8). Thus, the interfacial area covered by particles can be neglected for simplicity in the calculation. However, the constriction factor due to the reduced diffusion path should be considered in the calculations. As shown in Fig. 3b, the water diffuses across the oil layer driven by a concentration gradient of dissolved water vapor. It varies from $C_s$ at the water-oil interface to $C_o$ at the oil-air interface. For a transport-limited process, the value of $C_s$ can be considered as the solubility of water in oil. However, $C_o$ is unknown and is determined by assuming a steady-state process (refer to Supplementary Note 5 for detailed derivation). The temporal evolution of volume loss in the droplet is given by

$$\frac{dm}{dt} = -\frac{2\pi M_w D_o (C_S - C_\infty)}{\kappa} \frac{(\beta V^{1/3} + W)}{\left(\frac{W}{\beta V^{1/3}} + \frac{D_O}{D_A}\right)} \quad (10)$$

Here, $\beta^3 = 3[\pi(1 - \cos\theta_{LMOI})^2(2 + \cos\theta_{LMOI})]^{-1}$ where $\theta_{LMOI}$ is the contact angle of LMOI with the surface, *m* is the mass of the droplet, $M_w$ is the molecular weight of water (≈ 18 g mol⁻¹), $D_O$ is the diffusivity of water in oil (m² s⁻¹), $C_s$ is the solubility of water in oil (silicone oil ≈ 40 mol m⁻³,[36] mineral oil ≈ 2.95 mol m⁻³ [37,38]), $C_o$ is the concentration of water vapor at the oil-air interface, $C_\infty$ is the concentration of water vapor in the air (≈ 50% RH ≈ 0.567 mol m⁻³ at 25 °C), ρ is the density of water (≈ 997 kg m⁻³), *W* is the width of the oil layer (≈ 35 μm) and $D_A$ is the diffusivity of water in the air (≈ 3 × 10⁻⁵ m² s⁻¹), and $\kappa$ is the diffusion path correction factor (0 to 1).

Solving Eq. (10) numerically and fitting it with the experimental data suggests that the value of diffusivity of water in silicone oil ($D_O$) is ≈ 5.7 × 10⁻¹⁰ m² s⁻¹ and in mineral oil ≈ 9 × 10⁻¹⁰ m² s⁻¹ for the $\kappa$ = 1

(negligible constriction from particle). Even for the $\kappa$ = 0.2, the value of $D_O$≈1.1 × 10⁻¹⁰ m² s⁻¹ for silicone oil and 2.5 × 10⁻¹⁰ m² s⁻¹ for mineral oil. These values are in the same order as reported in previous literature (Fig. 3c)[39,40]. Interestingly, the diffusion coefficient of both mineral and silicone oil are in the same range; still, mineral oil-based LMOI has nearly 10 times a lifetime than silicone oil-based LMOI. It is because the solubility of water in mineral oil is significantly less than in silicone oil, which is responsible for an increase in a lifetime in mineral oil LMOI. The Eq. (10) is further tested for a bare droplet without any oil coating (*W* = 0 mm, $\kappa$ = 1 and $C_s$ = 1.134 mol m⁻³ at 25 °C), which results in a value of water diffusion in the air ($D_A$) as ≈ 3 × 10⁻⁵ m² s⁻¹, that is approximately the same as reported in the literature (Supplementary Fig. 15a)[41]. Thus, by choosing the appropriate particle size and oil with lower diffusivity and water solubility, the current limit of 12 days can be further increased.

## Effect of temperature
Most biological and chemical syntheses require higher temperatures to complete the process. Thus, the higher temperature stability of LMOI is also of utmost importance. Figure 3d shows the evaporation time of LMOI at various temperatures. According to the Stokes-Einstein relation, diffusivity $D \propto T \eta_o^{-1}$, where *T* is the temperature and $\eta_o$ is the viscosity of the oil. At higher temperatures, water diffusivity in oil increases due to an increase in temperature and a decrease in the oil viscosity. Thus, the droplet evaporates faster. Bubble evolution in oil at higher temperatures also plays a crucial role in deciding droplet lifetime. Bubbles disrupt the particle coating, and the oil film cannot be stabilized. Specifically, in mineral oil, bubble evolution at higher temperatures (> 90 °C) is very significant, probably because of the moisture content and lower bubble inception temperature, which decreases the lifetime significantly, as shown in Fig. 3d[42].

## Effect of oil viscosity
As predicted by the Stokes-Einstein relation, diffusivity should decrease as viscosity increases ($D \propto T\eta_o^{-1}$). Therefore, an increase in droplet lifetime should be observed with an increase in oil viscosity. Fig. 3e represents a similar outcome where up to 500 cP, an increase in lifetime is observed. The decrease in diffusivity with viscosity can be

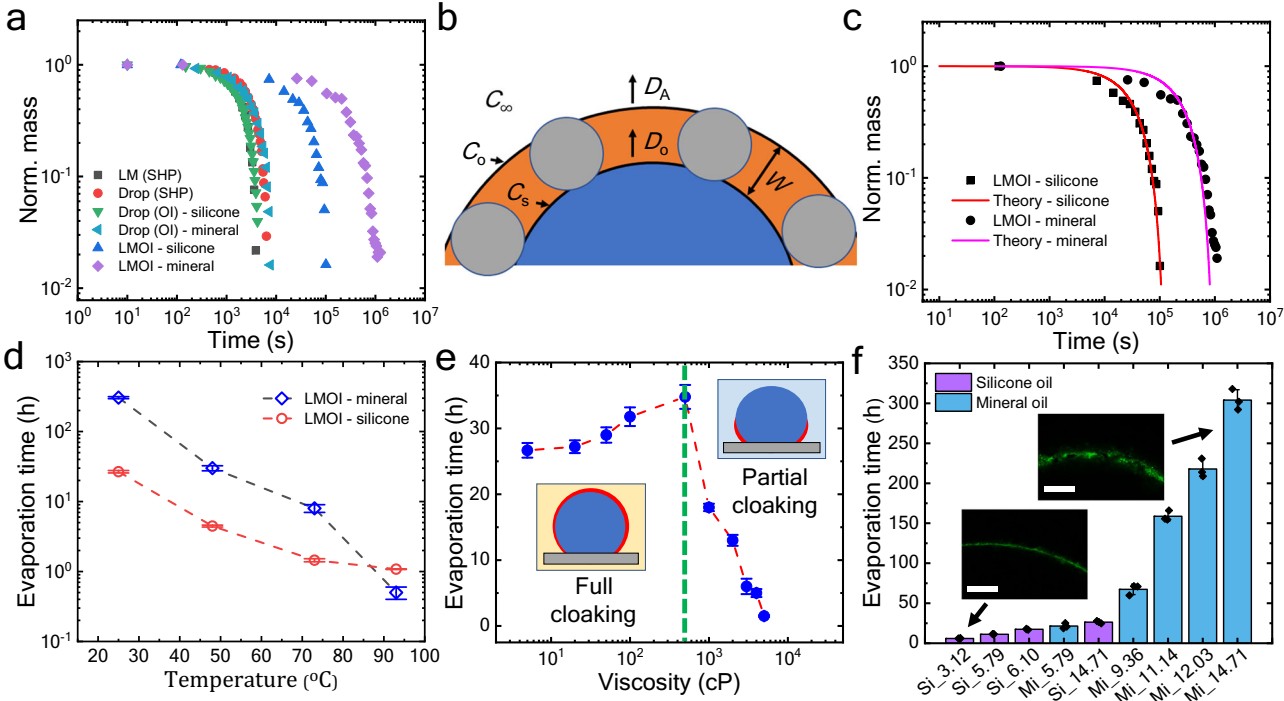

**Fig. 3 | Control of evaporation rate. a** Comparison of the temporal evolution of the normalized mass ($m/m_0$) of the droplet and LM in different configurations. Such configurations are droplet on SHP (red) and OI surface (silicone – green and mineral – cyan), LM on SHP surface (black), and LMOI based on silicone (blue) and mineral oil (magenta). The volume of the LM and droplet is 10 μL. All experiments were repeated $n = 3$ times independently, and the mean is plotted. **b** Schematic of LMOI defining various parameters important in the evaporation process where red color represents the oil layer and blue color region represents the water. **c** Comparison of the temporal evolution of normalized mass with the theoretical model. **d** High-temperature stabilization of 10 μL LMOI with mineral (blue color) and silicone (red color) oil. Measurements were carried out on $n = 3$ independent samples, and data are presented as mean values ± SD. **e** Effect of silicone oil

viscosity on evaporation time of the 10 μL LMOI. The green dotted line divides the full and partial cloaking of LM (inset for schematic) based on viscosity. Measurements were carried out on $n = 3$ independent samples, and data are presented as mean values ± SD. **f** Tunability of evaporation rate by changing the oil and particle coating density/mass loading. In the horizontal axis, Si and Mi prefixes indicate the LMOI with silicone oil (purple color) and mineral oil (blue color); the suffix number represents the mass loading of the particle on LMOI in μg mm$^{-2}$. Insets represent the confocal images of the different mass loaded LMOI. Scale bars = 200 μm. Measurements were carried out on $n = 3$ independent samples, and data are presented as mean values ± SD. Black dots represent individual data points. All experiments are performed at 25 ± 2 °C and (50 ± 3)% relative humidity. Source data are provided as a Source Data file.

further confirmed by Supplementary Fig. 15b. However, a further increase in viscosity results in a steep reduction in a lifetime. This behavior can be explained in terms of the time scale of cloaking. As per Eq. (9), the time required for the complete cloaking is given by $t = \tau_{LMOI} c^{-2}$. For lower viscosity (10 cP and pitch p = 100 nm) silicone oil, the $\tau \approx 10^2$ s, which is significantly shorter than the bare-droplet evaporation time scale ($\approx 10^3$ s). Thus, full cloaking is possible. However, for the high-viscosity oils ($\approx 5000$ cP), the cloaking time $\tau \approx 10^4$ s, which is one order of magnitude higher than the bare droplet evaporation time scale ($\approx 10^3$ s). Thus, the LM evaporates before it is fully covered by oil. Hence, a decrease in lifetime is observed.

**Effect of oil layer thickness**
Changing the particle size would change the cloaking layer thickness. Based on the model, the variation in evaporation time with respect to the particle size can be deduced. Supplementary Fig. 15c represents the tunability of the droplet's lifetime from the 1 μm to 200 μm range. Moreover, the lifetime can also be tuned by changing the mass loading of the particle coating. We have prepared a variable mass loading LM by coalescing LM with bare droplets with a final volume fixed at 10 μL (Supplementary Fig. 5a and Supplementary Table 2). This approach leads to variable surface fraction LM. The thickness of oil coating varies with the surface fraction (Inset Fig. 3f and Supplementary Fig. 5b, c). Such variation in oil coating thickness provides a variable evaporation rate and, thus, the variable lifetime of the droplet. By employing

variable coating, the lifetime of the droplet can be tuned from several hours to several days (Fig. 3f). Other methods of tuning the lifetime of LMOI and their resolutions are described in Supplementary Table 4. Moreover, Supplementary Table 5 compares the different techniques of evaporation prevention with the LMOI, where LMOI can be seen to outperform other methods.

**Single crystal growth in droplets**
Several applications, such as crystal growth and evaluating reaction kinetics, need a controlled evaporation of the droplets. Control of evaporation is of utmost importance in single crystal growth as it ensures the optimum growth rate. Previously reported methods, such as submerging LM or droplets inside an immiscible phase, cannot tune the evaporation rate[27]. Further, submerging is unsuitable for many applications as it segregates the droplet from the atmosphere and hinders basic operations such as gas exchange and mixing. In biochemical assays, external forces are necessary to merge submerged droplets[15]. In micro-bioreactors, submerged droplets require frequent changes in droplet liquid for successful cell growth[43]. In crystal growth, submerging causes very low evaporation rate (>1000 h lifetime); thus, the time taken to form a single crystal is very long[44]. Very slow evaporation creates problems in the initial screening and process optimization[45]. In detail comparison of the various techniques of single crystal growth has been enlisted in Supplementary Table 6.

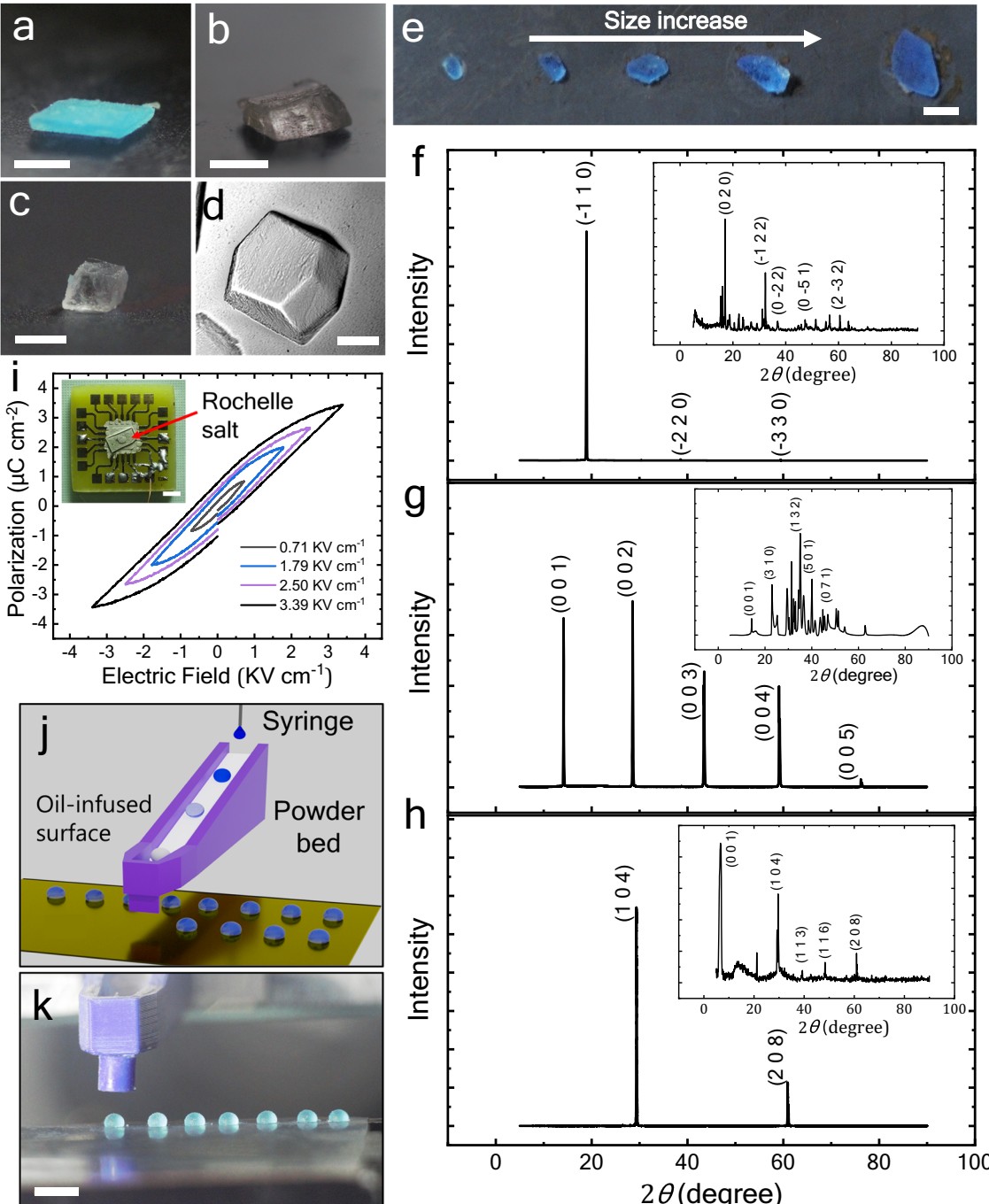

**Fig. 4 | Crystal growth in tunable LMOI liquid shell.** Photograph of the crystals made from LMOI. Scale bar = 1 mm. **a** Pentahydrate copper sulfate **b** Rochelle salt **c**, sodium nitrate, and **d**, lysozyme protein. For lysozyme, the Scale bar = 100 μm. **e** Variable size crystals of copper sulfate were prepared by changing the volume of LMOI from 5 to 60 μL. Scale bar = 1 mm. The XRD plot confirms a single crystal of **f**, pentahydrate copper sulfate, **g** Rochelle salt, and **h**, sodium nitrate crystals produced from LMOI. Insets represent XRD of the crystals produced from the droplet evaporation on OI, where the polycrystalline structure is evident. **i** Polarization-electric field (PE) hysteresis loop at 20 °C for the single crystal

Rochelle salt crystal produced from LMOI. The data confirms ferroelectric behavior of the Rochelle salt crystal. Inset: a Rochelle salt crystal attached to a printed circuit board (PCB) for PE measurement. Scale bar = 2 mm. **j** Schematic and **k**, photograph of the automated setup for the large number fabrication of LMOI. Where the droplet is formed via a syringe and then rolled onto a slant powder/particle bed. Due to the rolling of the droplet, the hydrophobic particle settles at the interface of the droplet and forms an LM. Then, the prepared LM falls gently on the oil-infused surface. The surface is mounted on the automated X-Y stage. Scale bar = 8 mm. Source data are provided as a Source Data file.

As shown in Fig. 4a, b, c, we demonstrate the growth of three different crystal types (i.e., copper sulfate, Rochelle salt, and sodium nitrate) inside LMOI. Apart from various salt crystals, we successfully produced a single crystal of lysozyme protein (Fig. 4d) inside LMOI

(with carbon soot-based oil-infused surfaces). The LMOI allows us to tune the evaporation rate to an optimum value, ensuring single-crystal growth. Additionally, uniform coating in LMOI ensures minimum variation in evaporation rate around the surface area of the droplets as

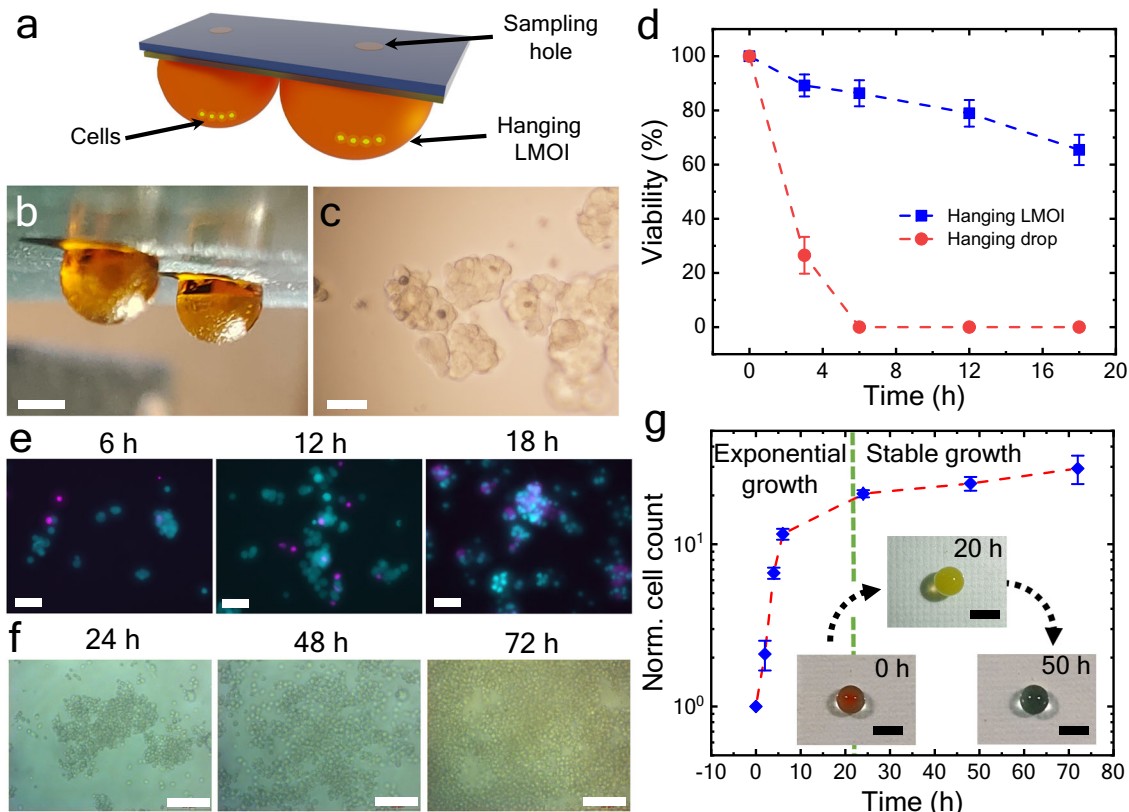

**Fig. 5 | LMOI as a biological reactor.** Hanging LMOI configuration as **a**, schematic and **b**, photograph. Scale bar = 1 mm. **c**, The cluster of ovarian cancer cells after 18 h of LMOI incubation at 37 °C and 5% CO₂. Scale bar = 50 μm. **d** Comparison of ovarian cancer cell viability in hanging LMOI (blue) and hanging droplet (red). Measurements were carried out on $n = 3$ independent samples, and data are presented as mean values ± SD. **e** Fluorescence photomicrographs were acquired at different time points for ovarian cancer cells, where cyan color represents live cells (due to staining with calcein acetoxymethyl ester), and magenta color represents the dead cells (due to staining with propidium iodide). Scale bar = 50 μm. **f** Yeast cell growth inside an LMOI at different time points. Scale bar = 50 μm. **g** Normalized yeast cell count variation with time where the number of yeast cells at a particular instance was normalized by the initial number of cells (at 0 h). The exponential growth can be seen for up to 20 h (green dotted line). However, after that, stabilization in growth is observed. Measurements were carried out on $n = 3$ independent samples, and data are presented as mean values ± SD. Inset: glucose levels of the yeast solution tested by Benedict's solution. Red, yellow, and green colors indicate glucose levels of more than 2%, 1%, and 0.5%, respectively. Scale bar = 1 mm. Source data are provided as a Source Data file.

variation in evaporation rate along different locations results in precipitation and fractal growth (Supplementary Note 9). The crystal size is controlled by changing the LMOI volume. Fig. 4e represents copper sulfate single crystals of different sized obtained by varying LMOI volume from 5 to 60 μL. The obtained crystals were analyzed with XRD, which confirms the single-crystal formation (Fig. 4f, g, h). In contrast, crystal growth within bare droplets results in polycrystalline structures due to fast evaporation (Inset of Fig. 4f, g, h). Additionally, the possibility of single crystal growth inside various drop-based methods with required time frames is given in Supplementary Table 7. Rochelle salt crystals are ferroelectric. To check the functionality of prepared crystals, we carried out a polarization-electric field (PE) hysteresis loop measurement on Rochelle salt (Fig. 4i), which confirms the ferroelectric behavior of Rochelle salt produced in LMOI.

The LMOI process can be scaled up easily to fabricate a large number of crystals. Figure 4j, k show the automated setup to fabricate the LMOI (Supplementary Movie 4). Supplementary Fig. 17a shows the fabrication of many single crystals based on LMOI with the help of the automated setup. A similar concept can be extended to create arrays of microreactors for various chemical and biological applications.

Allowing the LMOI to completely dry during crystal growth gives rise to crystals with particles attached to the surface (Supplementary Fig. 17b). This is undesirable for many applications. Thus, early harvesting of crystals is required so that particles do not stick to crystals. As shown in Supplementary Fig. 18, a small amount of oil is dispensed on the LMOI from the top. The oil flow drag takes the particle along,

allowing the LM to uncoat. Subsequently, the crystal is harvested. Such a method of particle uncoating also helps in tuning the lifetime of the LMOI.

## LMOI in biological applications

Contrary to droplets on oil-infused slippery surfaces, LMOI does not roll off once the oil covers the LM. The jamming of the particles at the three-phase contact line is the reason behind such strong adhesion. It stays stable even in hanging configuration, making it a useful alternative to the hanging drop methodology (Fig. 5a, b). Hanging LMOI can be used for the 3D culture of cells. Additionally, hanging LMOI possesses many advantages compared to its conventional counterparts (Supplementary Table 8). We investigated the formation of cell clusters by growing cells inside the hanging LMOI. Fig. 5c shows the 3D culture of ovarian cancer cells (OVCAR-3) inside the hanging LMOI. The cell clusters are formed after 18 h of incubation at 37 °C, 5% CO₂, and without any media reservoir.

Under similar conditions, hanging LMOI outperforms the hanging droplet configuration as cell viability is much higher in the case of hanging LMOI (Fig. 5d, e). Almost all cells died within the first 6 h in case of hanging droplets because of evaporation. In comparison, more than 65% of cells were alive even after 18 h in LMOI without replacing cell culture media. This can be improved with a mechanism to sample and replace media. The functionality of LMOI as a bioreactor is further tested by growing yeast cells. As shown in Fig. 5f, yeast cells were grown in an LMOI at room temperature without any humidity control.

As shown in Fig. 5g, the normalized cell count of yeast cells increases exponentially up to 20 h. However, after that, the growth slows, probably due to the depletion of nutrients in the culture[46].

Accessing the inner fluid in LMOI without disrupting the particle coating is challenging as cracks in LMOI are not self-healing. This is, however, necessary for replenishing nutrients and sampling liquids for analysis. We enable access to the fluid inside LMOI by making a hole in the nanostructured surface prior to oil infusion. The LMOI is stabilized on such surfaces, and the inner liquid can be sampled through the hole by using a syringe (Supplementary Fig. 19 and Supplementary Movie 5). Sampling was done from the LMOI with yeast culture. Benedict's test was performed on the sampled liquid to check the glucose level in yeast media. As shown in the Inset of Fig. 5g, the sampled liquid shows a drop in glucose level from more than 2% (red color) to less than 0.5% (green color) in 50 h, further confirming observations reported in Fig. 5g.

## Capsules

Solid-based encapsulation of the droplets has also been reported for many critical applications, such as the industrial formulation of pharmaceutical drugs and agricultural chemicals[21,22]. However, such encapsulations suffer mainly due to diffusion of the inner liquid, especially for porous polymeric shells[17,47]. Hard non-porous materials can be used for encapsulations; however, such encapsulations are difficult to engineer for stimuli-responsive release of the inner liquid. Recently, a paraffin wax-based encapsulation has been explored for hermetically sealed capsules with the ability to release material upon the application of temperature stimuli[16,17]. However, previously explored methods of preparing wax-coated capsules do not possess the ability to tune the thickness of the shell. The thickness tunability is desired for making capsules with variable strength and controlling the release time of the inner liquid. Detailed comparisons of various capsule-making techniques are given in Supplementary Table 9.

A particle-enhanced cloaking can be utilized for making thickness-controlled stimuli-responsive capsules. For this purpose, a temperature-responsive oil-infused surface based on paraffin wax is used. Below the melting point of wax ($T_m \approx 55\,°C$), the wax is solid, and the surface is hydrophobic (Fig. 6a); thus, the LM retains its shape. However, above melting temperature, wax is in a liquid state, which turns the surface slippery (Fig. 6b). Owing to lower viscosity at higher temperatures (5.84 mPa s at the coating temperature of 70 °C) (Supplementary Fig. 20), the molten wax driven by the capillary forces rises and cloaks the entire LM. After the complete cloaking of LM, the temperature is again reduced below $T_m$, solidifying the wax, and a solid capsule is produced. Half-cut capsule after complete drying is represented in Inset Fig. 6g. The wax infuses inside the porous structure of LM particles without displacing them.

The critical rupture pressure was measured using a motorized stage to press a capsule on a weighing balance (Supplementary Fig. 21). Fig. 6c shows the critical pressure for the breakage of the capsule with variation in particle size. Higher shell thickness offers a higher strength against rupture. A methylene blue (MB) encapsulated capsule was placed inside a water bath to test the diffusion of hydrophilic dye through the wax coating. Fig. 6d shows the absorption spectra of the water bath under different conditions. Even after two months of incubation, no traces of MB were found in a water bath (Fig. 6e), confirming the hermetic sealing of the inner liquid.

The prepared capsules show a stimuli-responsiveness to the temperature. As temperature increases, the wax layer on the capsule melts and releases the inner liquid (Fig. 6f). Moreover, the release time can be tuned by modulating the shell thickness. Fig. 6g shows the release time when the bath temperature is raised to 90 °C. An increase in capsule thickness results in a delay of rupture. The same is further shown in Fig. 6h, where two capsules with different shell thicknesses

were put in a hot water bath (Supplementary Movie 6). It is worth mentioning that the mechanism of delayed release is not driven by melting time difference. Both the shells melt within ≈ 100 ms. We believe that the difference in release time is due to the thinning of the molten wax shell under the influence of Marangoni flow inside the bath (Supplementary Note 13).

The substrate attachment of the capsules can be an issue for many applications. The substrate is removed by wet etching the nanostructured surface with an ammonium persulfate solution (Fig. 6i). This solution etches copper away but does not affect the wax capsule. Thus, LMOI-based solid encapsulation allows us to tune the shell thickness, strength, and on-demand release behavior.

## Optical and mechanical properties of LMOI

One of the crucial aspects of all the mentioned applications is the transparency of microreactors. It is well known that due to the PTFE particle layer, LM is opaque in nature (Fig. 7a)[48]. The roughness of the particle coating is responsible for scattering and, thus, less transparency. In comparison, oil infusion in the particle layer makes LMOI transparent. The oil layer reduces the scattering of light, and transparency is increased (Fig. 7b)[49]. Fig. 7c represents the transmission efficiency of LMOI compared to LM, where transparency of LMOI can be seen to increase by 2 to 3.5 times compared to LM.

Merging and mixing are two basic operations of any bio-chemical microreactor. Due to the particle and oil layer, it is difficult to merge two LMOIs. Fig. 7d represents the critical normalized deformation ($\Delta L/L$) of LMOI before merging. A significant amount of strain (>0.15) is required to merge two LMOI, while two bare droplets merge instantaneously as soon as they are brought into contact. Thus, LMOI has much higher mechanical stability compared to droplet-based reactors. By using the functional oils, the LMOI can be made stimuli-responsive. The LMOI prepared from the ferrofluid-infused surface (LMOI - ferrofluid) also shows responsiveness to the magnetic field. Fig. 7e represents the opening of LMOI upon application of a magnetic field (Supplementary Movie 7). Such on-demand opening helps in the removal of particle coating for accessing the inner solution. Similarly, magnetic field-based manipulation of LMOI can also help in merging LMOI on-demand. Fig. 7f reports the two LMOI-ferrofluid in contact with each other without merging. However, upon application of a magnetic field, the coating of LMOI ruptures, and merging happens (Supplementary Movie 7). Thus, the LMOI platform can also be implemented for the on-demand manipulation of droplet-based reactors. Additionally, the cloaking process is not limited to droplets only. The same process can be extended to low curvature and flat surfaces as well (Supplementary Notes 14 and Supplementary Movie 8).

In summary, we report a concept of LMOI, which results from the enhanced cloaking of a droplet. The presented technique overcomes the fundamental problem of flow-induced particle disruption in encapsulating particle-coated droplets using an immiscible liquid. The encapsulation by liquid and solid shells was demonstrated for 5 μm to 200 μm shell thickness with droplet volume ranging from 14 nL to 200 μL. Additionally, liquid shell encapsulation enhances the lifetime of 10 μL droplets up to 200 times with the tunability from 1.5 h to 12 days. Such an enhanced droplet lifetime enables cell growth and spheroidal culture applications such as multi-organ system-on-chip for drug discovery and personalized care. Thus, LMOI can substitute the hanging droplet method for many lab-on-chip applications without any special arrangements. Tunability of evaporation rate helps in growing single crystal materials. LMOI platform for crystal growth outperforms its conventional counterparts in terms of better stability, providing optimum growth conditions, initial screening, and easy process optimizations. Additionally, cloaking-assisted encapsulation is also demonstrated to be suitable for solid shells, thus, expanding its applications to chemical and material storage. Besides storage, the

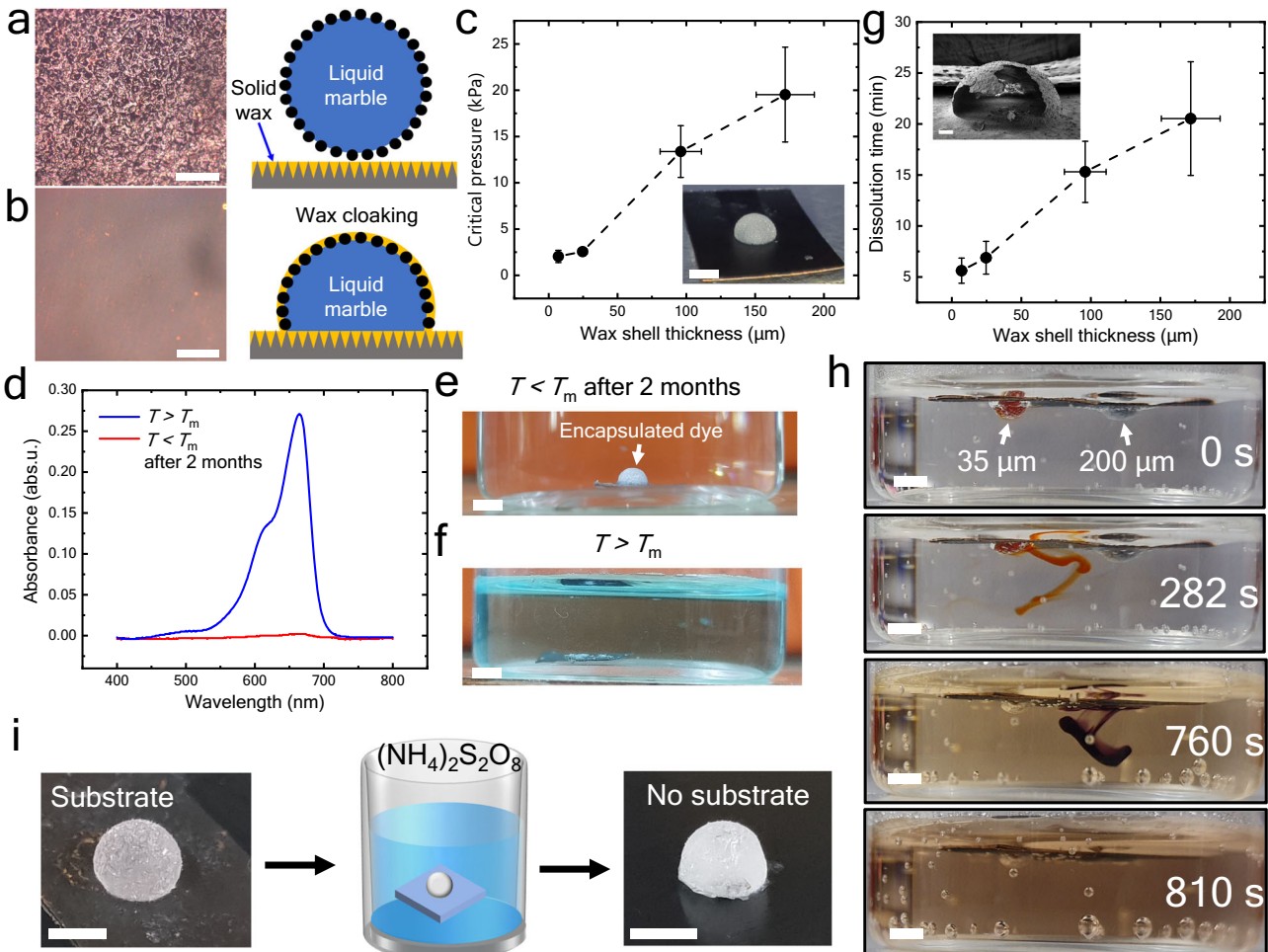

**Fig. 6 | Solid shell temperature-responsive encapsulation.** A wax-infused nanostructured surface and stabilization of LM at **a,** $T < T_m$ and **b,** $T > T_m$, Scale bar = 200 μm. **c** Required critical pressure to break the capsule. Inset: photograph of a capsule. Measurements were carried out on $n = 3$ independent samples, and data are presented as mean values ± SD. Scale bar= 2 mm. **d,** The absorbance spectra of the water in which the MB capsule was placed for **e,** 2 months at $T < T_m$ (blue color) and **f,** $T > T_m$ (red color). The absence of an absorbance signal at 664 nm suggests no diffusion of MB into water even after 2 months of immersion. Scale bar = 2 mm. **g** The time to dissolve the wax shell if the temperature of the fluid bath increases above melting temp ($T \approx 90\ °C$). Measurements were carried out on $n = 3$ independent samples, and data are presented as mean values ± SD. Inset: ruptured capsule under SEM. Scale bar = 200 μm. **h,** The tunability of release: two capsules with different shell thickness and filled with dye were placed in water at $T \approx 90\ °C$. Encapsulated dye placed in water where lower thickness (≈ 35 μm, red color) capsule ruptures early, releasing the inner liquid. In comparison, a higher thickness (≈ 200 μm, purple color) capsule ruptures after a prolonged duration. Scale bar = 2 mm. **i,** Removal of the capsule from the substrate by etching the surface in ammonium persulfate solution. Scale bar = 2 mm. Source data are provided as a Source Data file.

tunability of the solid shell helps in controlling its strength and on-demand release, extending the LMOI applications to drug and pesticide delivery. Scalable production of LMOI ensures its applicability for low-cost fabrication and industrial usage. The LMOI shell can be made stimuli-responsive using functional oils, which can be further utilized for various open-chip droplet manipulation platforms.

## Methods

### Liquid marble fabrication

A PTFE powder of 35 μm, purchased from Sigma-Aldrich, was used to prepare LM. The liquid droplet of the desired volume was gently rolled several times for the preparation of LM[50]. Control of the mass loading was obtained by fixing the initial volume of the LM and subsequently increasing its volume to 10 μL by merging it with the bare water droplet (Supplementary Fig. 5a)[51]. The initial volume for the LM preparation is determined by the geometric relation $S^3 \approx V^2$, Where $S$ and $V$ are the surface area and the volume of the liquid drop, respectively. The whole process was carried out on a superhydrophobic surface to prevent LM from breaking.

Apart from PTFE, LM can also be fabricated by other hydrophobic particles such as Lycopodium, Zein, and hydrophobic glass beads. Except for glass beads, all other particles were used as received without any further processing. The glass beads were hydrophobized with the help of Glaco mirror coat solution. The glass beads were first washed with acetone, isopropyl alcohol (IPA), and deionized (DI) water. Then, they were submerged inside the Glaco solution. After that, the entire solution was evaporated at 110 °C for 30 min, and hydrophobic glass beads were collected.

### Oil-infused surface fabrication

The nanostructured surface for the oil infusion was prepared from copper surfaces. First, the copper surface (3 cm × 2 cm) was cleaned with acetone, IPA, and DI water, followed by a 30 s cleaning with sulfuric acid. The cleaned copper surface was then immersed in an aqueous solution of sodium hydroxide (2.5 mol L⁻¹, 98% purity) and ammonium persulfate (0.1 mol L⁻¹, > 98% purity) for 20 min at room temperature. This solution etches the copper surface and produces nanowires on the surface of copper. The etched copper surface was then cleaned multiple

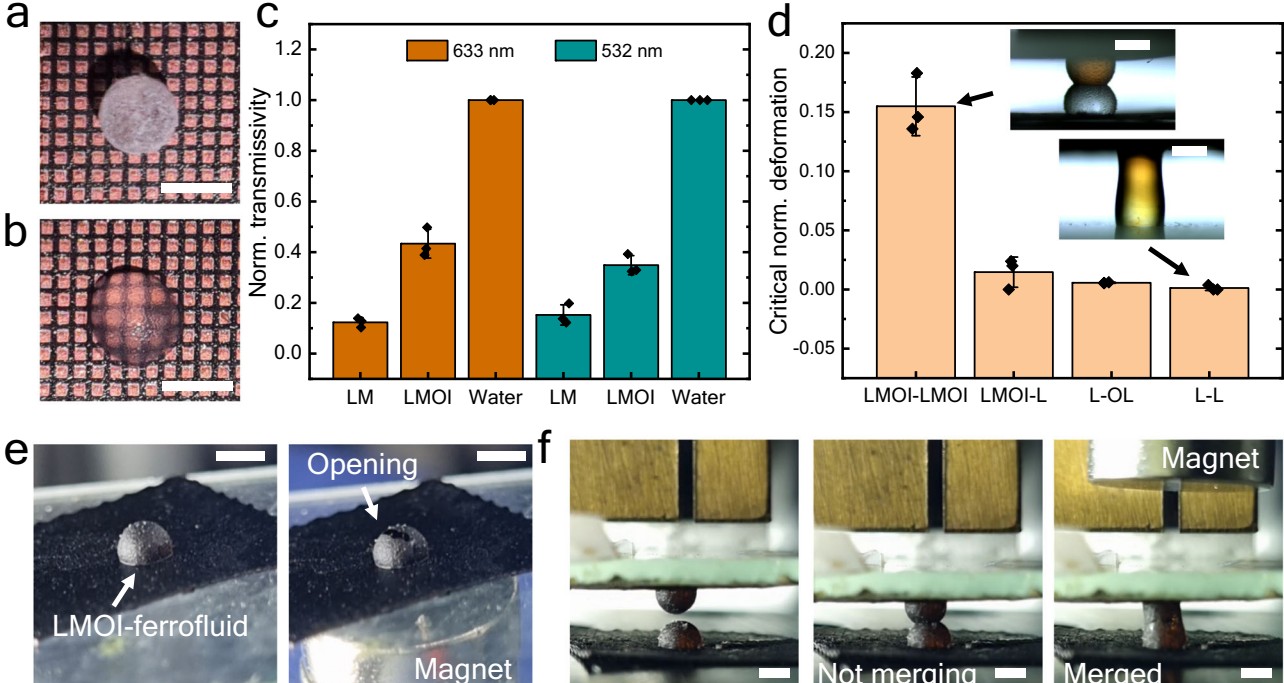

**Fig. 7 | Functionalities of the LMOI platform. a** LM on a structured surface where the bottom pattern is invisible. **b** Transparency of LMOI (silicone oil) where a clear pattern is visible through LMOI. Scale bar = 2 mm. **c** Comparison of transparency between water droplet, LMOI (silicone oil), and LM for two different laser wavelengths, i.e., 633 nm (dark orange) and 532 nm (dark cyan). Measurements were carried out on $n = 3$ independent samples, and data are presented as mean values ± SD. Black dots represent individual data points. **d** Critical normalized deformation ($\Delta L/L$) before merging for different configurations. L represents the bare liquid droplet, and OL represents the liquid droplet covered with an oil layer. LMOI (silicone oil) has been used throughout the experiments. Inset: merging photographs for different configurations. Measurements were carried out on $n = 3$ independent samples, and data are presented as mean values ± SD. Black dots represent individual data points. Scale bar = 2 mm. **e** Ferrofluid-based LMOI, where uncoating of the particle layer can be performed with the help of a magnet. Scale bar = 2 mm. **f**, Merging occurs when a magnet is brought closer to the non-merging LMOI − ferrofluid. Scale bar = 2 mm. Source data are provided as a Source Data file.

times with DI water and dried with nitrogen. Then the surface was dipped into the stearic acid solution in ethanol (5 % w/v) for 3 h. After that, the surface was rinsed multiple times with ethanol and dried with nitrogen. The prepared surface behaves as a superhydrophobic surface. However, it still allows oil to be infused. Several oil droplets were dropped down on the surface for oil infusion, and then nitrogen drying was performed to remove excess oil. The LM was then gently placed over the oil-infused surface for the preparation of LMOI. The etched surface can also be coated with Teflon instead of stearic acid. However, due to the high repellency of the Teflon-coated surface, higher surface tension liquids such as mineral oil are difficult to be infused in such surfaces. The stearic acid, sodium hydroxide, ammonium persulfate, paraffin wax, and silicone oil were purchased from Sigma-Aldrich. The paraffin mineral oil was purchased from SRL Chemicals.

The oil-infused surface can also be fabricated using soot particles deposited on the surface[52]. This methodology was specifically employed for protein crystallization due to the surface's non-reactivity in acidic environments. Initially, soot particles were deposited onto a pristine glass surface by directly positioning it above a wax candle. Subsequently, a mixture of polydimethylsiloxane (PDMS) and hexane (>96% purity) was created in a 1:1 ratio, supplemented with a 10% w/w crosslinker. The resulting solution was subsequently spin-coated onto a soot-coated glass surface. Once the deposition process was completed, the surface was subjected to a temperature of 110 °C for a duration of 3 h to facilitate the solidification of the PDMS. Upon achieving solidification, the surface was infused with the desired oils.

### LMOI preparation
First, the mass-loaded LM required for the experiment was prepared. To ensure that the LM with lower mass loading remains intact, it was handled with a superhydrophobic surface. Subsequently, the LM was carefully placed onto the desired oil-infused surface. It is advisable to wait for a few min before initiating any further procedures, allowing sufficient time for the oil to cloak the LM completely. It should be noted that the preparation of LMOI can be accomplished using various hydrophobic particles (Supplementary Fig. 24).

### Evaporation experiments
The evaporation experiments were performed in a custom-made sealed chamber. The humidity of the chamber was maintained at (50 ± 3)% at 25 ± 2 °C by using the saturated salt solution of magnesium nitrate unless conditions are specifically mentioned. The imaging of evaporating LMOI (10 μL) was captured via a camera. The subsequent image analysis was performed to find the instantaneous volume of LMOI. All experiments were repeated for 3 times independently and the mean was plotted.

### Confocal imaging
Coumarin 6 (98% purity), a hydrophobic fluorescent dye, was purchased from TCI. The oil was labeled with Coumarin 6 with a concentration of 50 μg g$^{-1}$ and sonicated for 1 h. The labeled oil was then infused into the nanostructured surface, and LMOI was prepared from it. The oil layer was visualized by a Zeiss LSM 880 microscope.

### Crystal Growth
The copper sulfate (99.5% purity), Rochelle salt (99% purity), and sodium nitrate (99% purity) were purchased from Sigma-Aldrich in a powder form. The solution was prepared by dissolving copper sulfate (0.3 g mL$^{-1}$), Rochelle salt (1.5 g mL$^{-1}$), and sodium nitrate (0.9 g mL$^{-1}$) in DI water. The desired volume droplet of such solution was then used

for preparing silicone oil based LMOI. The crystal structure was elucidated through rocking curve $\theta/2\theta$ parallel beam XRD measurements using a high-resolution monochromator source (Cu-K$\alpha$, $\lambda = 1.5406$ A°) in Rigaku Smart lab High-Resolution X-Ray Diffractometer. The authors referred to the ICDD database and VESTA software for all crystal studies for the powder diffraction database.

The protein crystals were prepared using lysozyme powder (egg white, Muramidase, molecular biology grade) obtained from SRL Chemicals. To create the crystal, an acetate buffer solution (0.1 M, pH 4.4) was prepared by combining sodium acetate (5 mg mL$^{-1}$, 99% purity) and acetic acid (5.7 mg mL$^{-1}$, >99 % purity). 6.5% w/v NaCl was then added to the acetate buffer as a precipitant. In a separate step, 75 mg mL$^{-1}$ of lysozyme powder was dissolved in DI water through gentle mixing. A 1:1 mixture of the lysozyme solution and the buffer solution was employed to generate an LM. To make an LMOI, the LM was subsequently stabilized over a carbon-soot-based oil-infused surface. It is important to note that a copper-based oil-infused surface is unsuitable for protein crystallization due to the reactivity between copper and the buffer.

## Polarization-electric loop measurement

The Rochelle salt single crystal was first prepared by slow evaporation of LMOI. Then the prepared crystal was thinned down to 357 µm with the help of mechanical polishing. The thinned crystal was pasted over a customized PCB with the help of silver paste to make connections for measurements (inset of Fig. 4g). The polarization-electric (PE) loop measurement was carried out at 1 kHz and 20 °C, which is below the Curie temperature of Rochelle salt ($T_c = 24$°C).

## Cell culture and early cluster formation

OVCAR-3, an ovarian cancer cell line obtained from ATCC, was used for early cluster formation. Early passage cells were thawed from frozen vials and cultured for two to three passages to relieve the cryogenic stress and attain healthy cellular morphology. At each passage, the cells were trypsinized at around 70% confluency and maintain a split ratio of 1:4; cells were reseeded in 60 mm dishes with 3 mL of media and kept inside the incubator at 37 °C and 5% $CO_2$. For spheroid culture, a cell suspension of density 85,000 cells mL$^{-1}$ was used to prepare LMOIs and hanging droplets, which were then kept inside an incubator and imaged at 3, 6, 12, and 18 h time points.

For assessing cell viability, Calcein acetomethoxy (AM) ester and propidium iodide (PI) staining was performed at every time point. Calcein AM is a non-fluorescent derivative of Calcein that gets transported across the membrane into live cells, where the intracellular esterases remove the AM group. Consequently, the molecule gets trapped inside, emitting a strong green fluorescence (excitation 495 nm, emission 515 nm) throughout the cell. Dead cells lack active esterases, thereby labeling affects only live cells. PI is a fluorescent agent that can only cross compromised or damaged cell membranes. Therefore, it predominantly enters dead cells and binds to DNA by intercalating between bases, emitting a strong red signal (excitation 493 nm, emission 636 nm) from the nucleus. For imaging the spheroids at different time points, brightfield, as well as epifluorescence microscopy, has been performed at 10x magnification.

## Yeast growth

Yeast cells were grown from baker's yeast. 10 mg of yeast and 60 mg of glucose were mixed in 1 mL of DI water. Mineral oil-based LMOI was prepared from 20 µL of the yeast cell solution. Imaging was performed at several intervals via an optical microscope.

## Liquid sampling from LMOI and Benedict's test

The hole was made on a clean copper surface by a 0.5 mm drill bead. Then the surface was etched and made superhydrophobic by the technique mentioned above. The mineral oil was infused in the prepared surface to make the oil-infused surface, and subsequently, the LMOI was prepared from yeast solution and placed at the center of the hole. The sampling ($\approx 1$ µL) was done via a syringe in a hanging LMOI configuration at different time intervals. After every sampling, the hole was filled with PTFE particles to avoid unwanted evaporation.

Benedict's reagent was purchased from OBA Chemie PVT. LTD. A 1 µL sample from LMOI was mixed with 2 µL Benedict's reagent on a Teflon-coated glass substrate. The sample was then heated at 70 °C for 1 min, and the color change was noted, which indicated the level of glucose in the sample. Red, yellow, and green colors indicate glucose levels of more than 2%, 1%, and 0.5%, respectively.

## Preparation of wax-infused surface and measurement of critical pressure

The copper surface was first etched with the technique mentioned above. Then the surface was dipped into a molten wax container and gently taken out. The surface was then placed at room temperature for some time until the wax solidifies. During capsule preparation, the surface was placed on a hot plate with a temperature above the melting point ($T_m$) of wax. After the liquefication of the wax, the LM was placed on the surface. After ensuring the complete cloaking of the wax on an LM, the hot plate temperature was brought down to room temperature to solidify the wax, resulting in a capsule.

The critical pressure for rupturing the capsule was determined by the in-house developed setup using a motorized stage, weighing balance, and high-speed camera (Supplementary Fig. 21). First, the capsule was placed on a weighing balance. Then, the squeezing plate, attached to the motorized stage, was brought down at a constant speed (50 µm s$^{-1}$) from the top. The high-speed camera was set up on the side to record the process. The reading from the weighing balance was recorded constantly during the process[53]. The final mass during the rupture process was taken into account to calculate the critical pressure. The pressure is calculated by $P_c = mgA^{-1}$, where $P_c$, $m$, $g$, and $A$ are the critical pressure of rupture, mass recorded by the weighing balance at rupture, gravitational acceleration $g$, and area of contact of the tablet with the surface, respectively.

## Dye diffusion measurement

A MB capsule was dropped into a water bath at room temperature and incubated for 2 months. The UV-vis-NIR spectrometer was used to check the concentration of dye in the water bath. The absorbance spectra was taken after correcting the background with DI water. The sample size was fixed at 3 mL. Similarly, after the thermal bursting of the capsule, the absorbance was measured, in which a peak at 664 nm is observed, confirming the release of MB. However, even after 2 months of incubation, no absorbance signal was observed, suggesting no diffusion of MB through the wax shell.

## Capsule release from the substrate

Wax capsules were removed from the substrate surface by etching the bottom copper surface. For this, a capsule with the substrate is dropped inside an ammonium persulfate solution (15 mg in 50 mL of DI water). After the complete dissolution of the substrate, the wax capsule was floating on the water. The released capsule was taken out by using a porous mesh.

## Transparency measurements

Transparency measurements were carried out using 532 and 633 nm laser sources. The laser source was first filtered through a natural density filter to reduce the intensity and then allowed to fall over the sample (droplet/LM/LMOI). A power meter was placed opposite to the laser to measure the transmission efficiency.

## Reporting summary

Further information on research design is available in the Nature Portfolio Reporting Summary linked to this article.

## Data availability

The data that support the findings of this study are available from the corresponding author upon request. Source data are provided in this paper.

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

## Acknowledgements

The authors thank Sayanta Goswami and Prof. Ambarish Ghosh for helping us in setting up the transparency experiments. The authors also thank Pooja Punetha, Upanya Khandelwal, and Prof. Rajeev Ranjan for their help in experiments related to ferroelectricity. The authors acknowledge the National Nanofabrication Centre and the Micro/Nano Characterization Facility at CeNSE, IISc, for providing us with the fabrication and characterization facility. All the authors thank the Department of Science and Technology and the Ministry of Education, Government of India, for financial support. P.S. is supported by DST Indo-Korea Grant. R.B. is supported by the Wellcome Trust/D.B.T. India Alliance Fellowship/Grant [IA/I/17/2/503312] and the Department of Biotechnology, India [B.T./909PR26526/G.E.T./119/92/2017]. R.L., S.M. and D.S. are grateful to Prime Minister's Research Fellowship for the research support.

## Author contributions

R.L. and S.N. contributed equally to this paper. R.L., S.N. and P.S. conceived the idea. R.L. and P.S. wrote the paper. R.L. and P.S. developed the theoretical model. R.L., S.N. and C.D.M. performed experiments related to the characterization of evaporation. S.M., R.L. and C.D.M. performed experiments and measurements related to cell culture. R.L. and S.N. conducted the yeast growth experiments. D.S., S.N. and R.L. performed the experiments and measurements related to crystal growth. R.L. and B.S.R. conducted experiments related to wax capsule formation and ferrofluid LMOI. P.S., P.N. and R.B. acquired the funding and supervised the project.

## Competing interests

The patent related to this paper has been awarded by the Indian patent office (Application no. 202241047688, Patent no. 445807, Submitted by Indian Institute of Science, Bangalore). R.L., S.N., C.D.M. and P.S. are the inventors of the patent. The remaining authors declare no competing interests.
