## [Peer Review File · Nature Communications]

Tunable Encapsulation of Sessile Droplets with Solid and Liquid ShellsReviewers' comments:

Reviewer #1 (Remarks to the Author):

This manuscript reports the sessile droplet encapsulation by utilizing wicking-assisted cloaking through usage of hydrophobic particles and liquid-infused surfaces. The proposed LMOI method enables formation of tunable liquid and solid shells on these sessile droplets, of which the evaporation rate can be reduced substantially. The authors performed detailed analysis on the condition for encapsulation and LMOI stability by varying the type of oil and particle properties, resulting in stable oil layer on the liquid marble. Also, the authors have shown that the oil layer thickness as well as the evaporation time can be fine controlled by adjusting the particle size and the oil type with varying level of water solubility and diffusivity. The theoretical model on evaporation rate and the time scale comparison between oil cloaking and oil supply through the nanostructured surface were thoroughly investigated. Furthermore, the application of this technique to single crystal and cell culture platform also confirms the broader utility and versatility. With that said, I believe that the tunable cloaking of droplet demonstrated in this work will be of significant interest to the surface science and microencapsulation community. Therefore, I recommend publication of this work in Nature Communications and hope that the following suggestion and comments help the authors improve their work.

1. In Figures 1e, it is suggested to add error bars (SD or SE). Also, the organization of the subfigures may need further improvement. The solid shell part is mentioned in two separate parts throughout the manuscript and seems redundant and out-of-place. It might be better to relocate Figure 1F-H to Figure 5 and combine them.
2. The authors calculate the value of Δ_LMOI for silicone oil and mineral oil and compare with Δ_ow to conclude that both silicone and mineral oil exhibit a stronger affinity to rise in the case of LM than the bare droplet. While this conclusion seems plausible, it also raises the question whether the difference in the value among Δ_LMOI for silicone oil (~ 0.34 microJ) and mineral oil (~ 0.235 microJ) has any physical meaning. Any comments or discussions will be helpful.
3. In line 188, two cosines are used consecutively. Is this correct? Also, in line 189 and 224, it might be better to define what θ_cr and θ_SHP is.
4. The authors mention in line 207-209 that the encapsulation thickness is proportional to the size of the particle and immediately continue the discussion by denoting that for smaller particles, the shell thickness is larger due to the agglomeration of the particles. This needs further clarification. How small is small?
5. In demonstration of the single crystal growth in droplets, does the directionality of evaporation have any affect on the size and shape of the resulting crystal?
6. What is the viscosity of paraffin wax used in this study at molten state? Unless the entire environment is carefully controlled, the interface facing the air phase might freeze and not cloak properly. Also, is the cloaking timescale smaller than the bare droplet evaporation time scale during paraffin wax-based encapsulation?
7. In line 481, typo in "Figure 1(F)", possibly "Figure 1(G)".
8. It is not clear why the two capsules with different shell thicknesses results in a delay of rupture. What is the mechanism of release?
9. In line 529, is the term, "missing" correct? Isn't it mixing?

Reviewer #2 (Remarks to the Author):

The manuscript describes an method to carefully coat a thin layer of oil onto liquid marbles. It could reduce the evaporation of liquid marbles, and several applications that were previously demonstrated by uncoated liquid marbles are re-demonstrated in this manuscript.

1. Overall this manuscript is not strong in innovation. Because the liquid marble (LM) covered with oil film was already reported by Ref. 28, which was also called composite LM. The only difference in Ref. 28 is that the droplet is covered by oil layer first and then covered by particles. Therefore, the thickness of the oil layer cannot be well controlled in Ref. 28. But I think the thickness of the oil layer can also be controlled in Ref. 28 by the amount of oil. The larger the amount of oil, the

thicker the oil layer will be formed.

2. The authors explored several potential applications of the composite LM, such as microreactor, capsule, crystallization and cell culture. But some of these applications are already not well established for liquid marbles, let alone oil-coated composite liquid marbles. For example, in figure 5, it was mentioned that the substrate attachment of capsules can be an issue for many applications. So the authors removed the substrate by wet etching with an ammonium persulfate solution, which is quite troublesome.

3. The merit of composite LM in this manuscript is the uniform oil layer as said in Line 155. But there is no experimental evidence.

Reviewer #3 (Remarks to the Author):

Rutvik et al reported a wicking-assisted cloaking of liquid droplet to produce uniform solid and liquid shell encapsulations (5-200 μm shell thickness for droplet volume spanning over four orders of magnitude). The tunable liquid encapsulation is shown to reduce the evaporation rate of tiny droplets by up to 200 times with a wide tunability in lifetime (1.5 hrs to 12 days). A few engineering demonstrations have been made to showcase the potentials. The topic is interesting, however, this manuscript didn't meet the merits to be published in Nat Comm.

The major concern is the lack of novelty. I guess one of the key science here is the cloaking mechanism of oil layer onto the LM, where the author tried to explain the understanding in Figure 1 and Figure 2. Apart from the observation and empirical summary, the scientific interpretation seems rather slim. It is obvious to me that the cloaking mechanism can be divided into the contact line of oil layer move 'through' the PTFE particle and 'over' the PTFE particle (from Figure 1H). Such transition of Liquid-particle interaction is unique from the corresponding macro-scale interactions due to the substantial role of capillary forces. The interaction of a single micro-particle with a dynamic contact line should be traced to uncover the underlying science. The three phase air-oil-LM contact line will be very helpful to elucidate the interaction mechanism and associated intertwined dynamics, including evolution and backward dragging of the transient air-oil-LM contact line, capillary-inertial launch of micro-particles and its subsequent trapping at the air-oil-LM contact line. Based on the force analysis, the author can provide a model to predict the interesting particle velocity profile and capillary-driven motion during the interaction. Unfortunately, this part of assessment seems totally missing, lead to an incremental scientific advancement in this work.

In the demonstrations, the authors did not describe what we have learned from this work about general material design/system setting and how we can use these. Even a series of demonstrations have been made, such as crystal growth, biological reactor, and functional platform, limited attractive/innovative points are presented, not mention that these demonstrations are well developed in other liquid droplet related research. May be author can start to think from some related questions as following, why did the authors choose the engineering concepts? How these concepts compared to the conventional technologies? How about the robustness of each application demo? ...The chemical - physical assessments on the showcased demo seem incomplete, which need a substantial effort to work on it in a systematic way.

In conclusion, the state of manuscript is pre-mature, which certainly doesn't meet the merits for publishing in Nat Comm. It might fit to some specified journal i.e. Sci. rep. etc. I would recommend to reject or transfer. However, I suggest authors to revise the manuscript based on above comments prior to submit to other journals.

Reviewer #4 (Remarks to the Author):

Summary: This manuscript describes a novel wicking-assisted method of cloaking droplets, its

detailed experimental demonstration along with theoretical analysis. Overall, the manuscript is interesting, however, some key components involved in the microparticles design are not sufficiently described or analyzed. It is not too clear what exactly is the groundbreaking novelty of the paper compared to the prior art. The paper should be reconsidered after a major revision, in which the authors clarify the importance of their findings and address the following critical questions.

Major Comments/Questions:

1. A detailed discussion is needed regarding the status of PTFE particles on the surface of a droplet, including the immersed percentage of particles, their dispersity (i.e., are they monodispersed on the droplet surface?), the density (i.e., the distance between nearby particles)? The density of the PTFE particles should be measured either by optical means or statistical methods such as checking the mass change post-particle coating.
2. How the above parameters would influence the stability of liquid encapsulation? For example, how does the density of microparticles influence the cloaking thickness and droplet stability?
3. Why were PTFE particles chosen for this method? Is the method generally applicable to other types of hydrophobic microparticles? This requires discussion.
4. Figure 5C: how can one explain the two different slopes for the pressure before breakage? What is their origin? Can these two different particle size regimes have different stabilizing mechanisms against mechanical stress?
5. The lifetime tunability of 10 μ L droplets is reported between 1.5 hours to 12 days, what is the resolution of the tunability here - can you program it by the hour, by minutes? Please provide some data and discussion.
6. The figures need to be made clearer by reducing the number of subfigures (ones that are not critical to the main point of the paper can be moved to the SI - such as Fig. 4H, for example).
7. Figure 5 uses the word "stimuli-responsive" but heating to melt the encapsulation is typically not considered a response to a stimulus within active/responsive materials community. Please reword this.
8. Figure 3F should be compared to a control experiment to demonstrate that the Rochelle salt property is only achieved by the method used by the authors.

Minor comments:

1. Figure 3H demonstrates scalability, but it is not the critical point of this work and can be moved to SI.
2. Please correct "cps" to cP in many instances throughout the manuscript

Response to reviewers' comments

Manuscript ID: NCOMMS-23-07654

“Tuneable Cloaking of Sessile Droplets”

We thank the Editor and reviewers for carefully reading our manuscript and providing their insightful comments. We believe these comments will improve the overall quality of the manuscript. Here, we provide a point-to-point response to these comments. For clarity, the reviewers' comments are in blue, and the changes we have made are highlighted in yellow in the revised manuscript.

Reviewer #1 Comments:

This manuscript reports the sessile droplet encapsulation by utilizing wicking-assisted cloaking through usage of hydrophobic particles and liquid-infused surfaces. The proposed LMOI method enables formation of tunable liquid and solid shells on these sessile droplets, of which the evaporation rate can be reduced substantially. The authors performed detailed analysis on the condition for encapsulation and LMOI stability by varying the type of oil and particle properties, resulting in stable oil layer on the liquid marble. Also, the authors have shown that the oil layer thickness as well as the evaporation time can be fine controlled by adjusting the particle size and the oil type with varying level of water solubility and diffusivity. The theoretical model on evaporation rate and the time scale comparison between oil cloaking and oil supply through the nanostructured surface were thoroughly investigated. Furthermore, the application of this technique to single crystal and cell culture platform also confirms the broader utility and versatility. With that said, I believe that the tunable cloaking of droplet demonstrated in this work will be of significant interest to the surface science and microencapsulation

community. Therefore, I recommend publication of this work in Nature Communications and hope that the following suggestion and comments help the authors improve their work.

1. In Figures 1e, it is suggested to add error bars (SD or SE). Also, the organization of the subfigures may need further improvement. The solid shell part is mentioned in two separate parts throughout the manuscript and seems redundant and out-of-place. It might be better to relocate Figure 1F-H to Figure 5 and combine them.

Response:

We thank the reviewer for the valuable feedback. We now have added the error bars (SD) in Figure 1D (Reproduced here as Figure 1).

Figure 1: Stability of LMOI for different oils. The negative value of $\gamma_{ow} - \gamma_o$ represents the unstable coating where crack formation in LMOI is present (Inset: LMOI – neem oil). The positive value of $\gamma_{ow} - \gamma_o$ represents the stable coating where no crack formation in LMOI is observed (Inset: LMOI – mineral oil). Scale bar = 1 mm.

The solid shell has been introduced in the initial part to provide additional support to shell thickness tunability. We have measured the shell thickness using the optical (confocal) technique. However, the resolution of our optical technique is limited due to the droplet shape

and depth-of-focusing limitations. Hence, we rely on solid-shell encapsulation to provide better shell-thickness data. Further, SEM images of the solid shell provide better insight into the structure of the shell, which is necessary for the development of the theory. We believe presenting these pieces of information in the early part of the manuscript will help the reader to better appreciate the subsequent parts. Relocating Figure 1F-H could obscure important structural aspects of the encapsulation and may create confusion for readers. Furthermore, Figure 5 illustrates a different aspect of our study, i.e., the characterization and functionality of solid shells.

2. The authors calculate the value of Δ_{LMOI} for silicone oil and mineral oil and compare with Δ_{ow} to conclude that both silicone and mineral oil exhibit a stronger affinity to rise in the case of LM than the bare droplet. While this conclusion seems plausible, it also raises the question whether the difference in the value among Δ_{LMOI} for silicone oil (~0.34 microJ) and mineral oil (~0.235 microJ) has any physical meaning. Any comments or discussions will be helpful.

Response:

The Δ_{LMOI} is defined as the energy difference between the cloaked and uncloaked liquid marble. It has a similar physical meaning as the spreading coefficient, as it can be defined as the $\Delta_{LMOI} = S_{LMOI} \times A_{surface}$ where S_{LMOI} is the spreading coefficient of oil on liquid marble and $A_{surface}$ is the surface area of the LMOI. When comparing the spreading behavior of two oils on the same LM surface, if one liquid has a more positive Δ_{LMOI} than the other, it will have a greater tendency to wet the LM surface and spread out (or rise) more readily. This is because of the larger capillary force (F_c) associated with large difference in Δ_{LMOI} as $F_c = -dE/dy$. However, it is very difficult to verify the difference in rise time as viscosity also affects the rise

time. Detailed model of the rising dynamics is given in “Dynamics of cloaking” section of the main manuscript.

It is worth noting that we revised our energy model with a more comprehensive version, which accounts for the contact angle change accompanied with the oil-rise. This has slightly changed the values of Δ_{LMOI} (Silicone oil - $\Delta_{LMOI} \sim 0.351 \mu J$ and mineral oil - $\Delta_{LMOI} \sim 0.286 \mu J$). We have added the following sentence in the main manuscript for further clarification.

“Liquids with higher positive Δ_{LMOI} values spread (or rise) more easily on the LM surface compared to those with lower Δ_{LMOI} values. This should result in a stronger driving force for cloaking and a faster spreading (rise) over the LM surface. Due to its higher Δ_{LMOI} value, silicone oil has a greater affinity for cloaking the LM surface than mineral oil.”

3. In line 188, two cosines are used consecutively. Is this correct? Also, in line 189 and 224, it might be better to define what θ_{cr} and θ_{SHP} is.

Response:

We thank the reviewer for pointing out this error. Only one cosine is present, which has now been corrected in the main manuscript.

θ_{LMOI} is the apparent contact angle of the LMOI on the oil infused surface (Figure 2). The critical contact angle, θ_{cr} , determines whether a stable LMOI will form on the surface.

Figure 2: Schematic representation of the apparent contact angle of the LMOI on the oil-infused surface

The following sentences have been added to the main manuscript for clarity.

“ θ_{cr} is defined as the critical LMOI contact angle that determines whether the LMOI will remain stable and crack-free on the surface or not.”

“The θ_{SHP} is the contact angle of the water droplet over the prepared superhydrophobic (SHP) surface.”

4. The authors mention in line 207-209 that the encapsulation thickness is proportional to the size of the particle and immediately continue the discussion by denoting that for smaller particles, the shell thickness is larger due to the agglomeration of the particles. This needs further clarification. How small is small?

Response:

The particles used in our study agglomerate below 1 μm in size, which was also reflected in the thickness data shown in Manuscript Figure 1(H). To investigate this further, we prepared wax capsules using 35 μm PTFE particles (non-spherical with a range of 25 μm to 45 μm) and examined them using SEM. The resulting image, presented in Figure 3(A), showed a thickness of around 25 μm , which is less than the average particle size, indicating the absence of agglomeration. However, when we prepared wax capsules using 800 nm particles, the resulting

thickness was around $6\ \mu\text{m}$, strongly indicating the presence of agglomeration (Figure 3(B)). This finding was further supported when we collected $800\ \text{nm}$ particles from the water interface onto a glass substrate. This was done using the Langmuir-Blodgett method (Figure 3(C)). A layer of $800\ \text{nm}$ particles was formed on a liquid bath. Subsequently, the particle film was lifted from the bath on a glass substrate. SEM image of the lifted particle film shows agglomeration (Figure 3(D)). The above explanation is added to the supplementary information of the paper for better clarity.

Figure 3: (A) Thickness of the wax capsule for $35\ \mu\text{m}$ particle size. Scale bar = $20\ \mu\text{m}$. (B) The thickness of the wax capsule for $800\ \text{nm}$ particle size. Scale bar = $10\ \mu\text{m}$. (C) Langmuir-Blodgett method, where particles were first stabilized on a water bath and then transferred onto a solid substrate by carefully dipping the substrate into the liquid surface. (D) Agglomeration size for $800\ \text{nm}$ particles lifted on a glass slide using the Langmuir-Blodgett method. Scale bar = $2\ \mu\text{m}$.

5. In demonstration of the single crystal growth in droplets, does the directionality of evaporation have any affect on the size and shape of the resulting crystal?

Response:

Depending on the contact angle, the evaporation rate near the contact line is known to be different from the rest of the droplet surface. This is because vapor diffusion in air dominates

the process. However, in our case, the evaporation rate is significantly lowered due to the oil layer. This is due to the very low saturation concentration of water in oil. This saturation-dominated slow evaporation rate significantly reduces (eliminates) the directional differences in the evaporation rate for the reported encapsulation scheme.

However, this will not be true for conventional oil covered compound droplets. We performed an experiment where a copper sulfate solution droplet was covered with silicone oil in a conventional compound droplet configuration. Here most of the oil is at the bottom near the contact line. The top of the droplet is covered with a very thin sub-micron thickness oil layer. Such a setup ensures minimal evaporation from the side while a higher evaporation rate from the top (Figure 4). This configuration resulted in faster evaporation, and single-crystal formation was not observed. We see polycrystalline fractal-like dendrite growth similar to water droplet freezing on oil-infused surfaces.¹ Thus, simple oil droplet-based encapsulation is not suitable for single crystal growth. The above explanation is added to the supplementary information of the paper for better clarity.

Figure 4: Temporal evolution of the copper sulfate solution covered by silicone oil on a Teflon-coated glass surface. Scale = 1 mm.

6. What is the viscosity of paraffin wax used in this study at molten state? Unless the entire environment is carefully controlled, the interface facing the air phase might freeze and not cloak properly. Also, is the cloaking timescale smaller than the bare droplet evaporation time scale during paraffin wax-based encapsulation?

Response:

Figure 5: The variation of paraffin wax viscosity with temperature.

The melting temperature of the paraffin wax (Sigma-Aldrich) used in our experiments is 55 °C. The viscosity variation of the paraffin wax is represented in Figure 5. The viscosity measurement was carried out in an Anton Paar rheometer with a linearly changing shear rate from 1 s^{-1} to 100 s^{-1} in a cone plate configuration. Even close to the melting point ($\sim 60 \text{ }^\circ\text{C}$), we find a very low viscosity of wax ($\sim 7.02 \text{ mPa}\cdot\text{s}$). At $70 \text{ }^\circ\text{C}$, the viscosity of paraffin wax is around $5.84 \text{ mPa}\cdot\text{s}$. Thus, in order to ensure proper cloaking, the wax-infused surface was heated at $70 \text{ }^\circ\text{C}$, and then LM was placed over it. At $70 \text{ }^\circ\text{C}$, the cloaking happens within the span of 1.5 min, which is around 15 times less than the droplet evaporation time at the same temperature. Thus, the molten wax coats the LM before significant evaporation. The time scale for heat diffusion inside a droplet is given by $t_{diff} \sim D^2/D_t$ where D and D_t are the diameter of the droplet and the thermal diffusivity of the inner liquid. Taking the thermal diffusivity of water as $0.145 \text{ mm}^2/\text{s}$ with a droplet diameter of 2 mm, $t_{diff} \sim 30 \text{ s}$, which is of the same order of magnitude as wax cloaking time ($\sim 90 \text{ s}$).² Thus, the droplet achieves an equilibrium temperature before the complete cloaking. This ensures very little to no thermal gradient for the wax to freeze. Since the cloaking temperature ($70 \text{ }^\circ\text{C}$) is well above the melting point of wax ($55 \text{ }^\circ\text{C}$), the wax layer is expected to be in the molten state even with minor fluctuations

in temperature. Additionally, we only observed the solidification when the whole setup was brought down to room temperature. The above explanation is added to the supplementary information of the paper for better clarity.

7. In line 481, typo in “Figure 1(F)”, possibly “Figure 1(G)”.

Response:

We thank the reviewer for pointing out this mistake. We have corrected it in the main manuscript.

8. It is not clear why the two capsules with different shell thicknesses results in a delay of rupture. What is the mechanism of release?

Response:

The release mechanism of the capsule depends on the melting of the wax shell. All the capsules were kept inside a water bath at the same time after reaching the bath temperature of 90 °C. The temporal evolution of the melting front is given by $L(t) \sim (\alpha_l t)^{0.5}$, where L , α_l , and t are the length of the melting front, thermal diffusivity in the liquid phase, and time instance, respectively.³ Taking $\alpha_l \sim 0.19 \text{ mm}^2/\text{s}$ ⁴ and thickness of capsule as $L \sim 200 \mu\text{m}$, the time for complete melting is in the order of 0.1 s, However, actual time is around 4 orders of magnitude higher ($\sim 10^3 \text{ s}$) than the predicted by melting front equation. Thus, in our case, the mechanism of capsule disintegration is very different from the melting based disintegration. We believe that the disintegration is the consequence of thinning of capsule wall under the influence of Marangoni flows in the water bath (Figure 6(A)). The shear of molten wax against the Marangoni flows results in thinning of the capsule and eventually disintegration. Since the shear thinning depends on the thickness of the shell wall, we observe increase in disintegration time with increase in thickness (Figure 6(B)). However, the exact modelling of the

phenomenon is complex and out of scope for this paper. The above details are added to the supplementary file.

Figure 6: (A) Schematics representation of capsule thinning through the Marangoni flow. (B) The time to dissolve the wax shell if the temperature of the fluid bath increases above melting temp ($T \sim 90^\circ\text{C}$).

9. In line 529, is the term, “missing” correct? Isn’t it mixing?

Response:

We thank the reviewer for pointing out this mistake. We have corrected it in the main manuscript.

Reviewer #2 Comments:

The manuscript describes an method to carefully coat a thin layer of oil onto liquid marbles. It could reduce the evaporation of liquid marbles, and several applications that were previously demonstrated by uncoated liquid marbles are re-demonstrated in this manuscript.

1. Overall this manuscript is not strong in innovation. Because the liquid marble (LM) covered with oil film was already reported by Ref. 28, which was also called composite LM. The only difference in Ref. 28 is that the droplet is covered by oil layer first and then covered by particles. Therefore, the thickness of the oil layer cannot be well controlled in Ref. 28. But I think the thickness of the oil layer can also be controlled in Ref. 28 by the amount of oil. The larger the amount of oil, the thicker the oil layer will be formed.

Response:

We thank the reviewer for taking the time to review our manuscript and for providing us with valuable feedback. While it is true that Ref. 28 reported on the use of an oil film in combination with particles to cover a droplet, our work differs from theirs in several important ways. Unlike other techniques, this report coats the particles first, and then an oil layer encapsulation is created. This approach has been attempted by other reports before. However, successful encapsulation was not achieved. This is due to a lack of control over the flow velocity of the encapsulating oil film. By solving this problem, we demonstrate an encapsulation technique that has several benefits over the composite LM (Ref. 28) technique, as shown in Table 1. We believe that our technique offers a significant improvement over the previous report. In the subsequent discussion, we provide experimental and theoretical evidence to show why it is not possible to obtain a uniform and tunable encapsulation using previously reported techniques.

	Composite LM	LMOI (This work)
Uniform shell	×	✓
Tunability	×	✓
Lifetime	< 3 hrs	>12 days
High-Temperature stability	×	✓
Hanging configuration	×	✓
Liquid sampling without disruption	×	✓
Single crystal growth	×	✓
Cell growth	×	✓
Particle removal on-demand	×	✓
High mechanical stability	✓	✓
Transparency	× (because of clumps)	✓
Solid capsule formation	×	✓
Stimuli-responsiveness	×	✓

Table 1: Comparison of composite LM with LMOI.

The technique presented in Ref. 28

In Ref. 28, the oil layer was spin-coated on a surface, and then a water droplet was slid over it. The sliding water droplet gets covered with a thin layer of oil. The oil-coated droplet is detached from the surface and then rolled over a particle bed. This makes a composite LM structure with a combination of oil and particles. Ref. 28 did not demonstrate thickness variation. A 2.4 μm oil thickness was estimated by carefully weighing the composite LM. In comparison, the method reported in this manuscript can achieve a wider range of oil thickness starting from 5 μm to 200 μm .

What if we used more oil?

It can be argued that a thicker encapsulation should be possible by using larger oil volumes for pre-coating the water droplet. However, this approach is not feasible as the oil coating on the droplet becomes non-uniform. Figure 7 shows the effect of using higher oil volumes for coating a $10\ \mu\text{L}$ water droplet. It is known that the balance of surface and gravitational forces leads to a non-uniform oil layer over the droplet (see Figure 7).⁵⁻⁸ Further, during the sliding and detachment of the droplet from the surface, viscosity and density differences will also contribute to the non-uniformity in the oil layer thickness. The non-uniformity in oil layer thickness results in a variable particle coating.

Figure 7: The sessile water droplet ($10\ \mu\text{L}$) covered with different amounts of silicone oil (20 cP) on a Teflon-coated glass. Scale = 0.5 mm.

Timescale of oil infiltration in the particle bed

Apart from the oil encapsulation reported in Ref. 28, it is also very difficult to get a uniform coating of oil over a droplet with other methods. Such methods include concentric

nozzle/needle, where the core liquid can be ejected from the central needle, and shell material can be injected through the outer needle. However, the uniformity of the shell is still very hard to achieve and limited to some special liquids only. Even a slight variation in the density between core and shell liquid can significantly impact the mechanical stability of such droplets.^{9,10} When a compound drop, consisting of a core phase and a surrounding shell phase, descends through the air, the core phase tends to accelerate downward if it has a higher density compared to the shell phase. Conversely, if the core phase is less dense, it tends to move upward. Thus, the above problem forces the needle-based systems to have a density match between core and shell liquids and thereby limiting the choices of liquids.¹¹

In addition to the challenges posed by the non-uniformity in coating oil before particles, the method described above also encounters issues with particle clumping, which introduces additional non-uniformity in the coating process. When oil-coated droplets impact the particle bed, the oil has a tendency to seep through the bed, resulting in the formation of clumps or agglomerations at the interface where the oil thickness exceeds the size of the particles. These clumps significantly contribute to the non-uniformity of the coating, as depicted in Figure 8. The timescale of wetting in porous media is governed by the Washburn equation if $\eta \gg \rho^{\frac{3}{2}} g R^{\frac{5}{2}} / \gamma^{\frac{1}{2}}$.¹² For $\rho \sim 1000 \text{ kg/m}^3$, $R \sim 35 \text{ }\mu\text{m}$ and $\gamma \sim 20 \text{ mN/m}$, $\rho^{\frac{3}{2}} g R^{\frac{5}{2}} / \gamma^{\frac{1}{2}} \sim 10^{-5} \text{ Pa}\cdot\text{s}$ which is around two to four orders of magnitude lower than the viscosity used ($\eta \sim 10^{-3} - 10^{-1} \text{ Pa}\cdot\text{s}$) in the present paper; thus, the Washburn equation is valid in our case as well.

According to the Washburn equation, the time of wetting is given by $\tau = 4\eta l^2 / \gamma D$, where l is the wetting length (\sim two particle layers for clump formation $\sim 70 \text{ }\mu\text{m}$) and D_p is the diameter of the particle ($35 \text{ }\mu\text{m}$). Thus, assuming clumps of particles occur as wetting takes place over two layers, the minimum time required for clump formation is estimated as $\sim 0.28 \text{ ms}$. The

timescale of contact for the impact of such a droplet is in the order of 10 ms, which is 2 orders of magnitude higher than the clump formation time scale. Therefore, in a very short period of time, the multiple layers of particles in the particle bed get wetted and adhere to the LM surface, which causes non-uniform clumps all around the composite LM. This also explains the reason behind having a higher oil layer thickness compared to particle size in Ref. 28. Thus, particle coating has to be done before oil coating to ensure uniform and tunable coating throughout the interface.

Figure 8: The composite LM prepared by the technique mentioned in Ref. 28 by using silicone oil with $35 \mu\text{m}$ particle size. Scale bar = 2 mm.

Problems with agglomeration (clumping)

We evaluated the effect of clumping on the evaporation rate. Figure 9 represents the normalized mass evolution for the LMOI prepared by the technique reported in this paper and Ref. 28 for $10 \mu\text{L}$ droplet. The composite LM was prepared by using $0.8 \mu\text{L}$ mineral oil to have nearly $35 \mu\text{m}$ thickness. The LMOI lifetime is around two orders of magnitude higher than Ref. 28 lifetime. This is because of the uneven coating of the composite LM (Ref. 28) as vapor escapes through the thinly coated sides, while in the case of LMOI, the uniform coating ensures no abrupt escape of vapor.

Other limitations of the technique presented in Ref 28.

Other than non-uniformity, it is very difficult to form a solid encapsulation with the composite LM. The wax coating prior to particle coating, faces similar problems as coating oil. In addition, the wax requires a high-temperature setup where all process has to be carried out at a high temperature, which is not feasible for lower-volume droplets. Moreover, dropping a wax-coated droplet over a particle bed may solidify the wax in the air before it touches the bed.

Figure 9: The normalized mass evolution for the LMOI prepared by the proposed technique and Ref. 28 for 10 μ L droplet.

Why it is not possible to coat LM by simple dispersal of oil on LM?

Further, the same work (Ref. 28) demonstrated that coating particles before oil exposure is not trivial as it displaces the particles and ruptures the particle coating (See Supplementary Video S1). When a liquid marble (LM) is exposed to oil from the top, the capillary forces drive the oil to encapsulate the liquid marble. The hydrodynamic drag force on the particle scale as $\propto \eta D_p v$, where η is the oil viscosity, D_p is particle diameter, and v is the capillary rise velocity of oil. The oil encapsulation velocity v is observed to be ~ 10 mm/s. Thus, hydrodynamic drag on 35 μ m particles and a 10 cst oil is in the order of $\sim 10^{-8}$ N. The adhesive force on a particle is given by $\propto \gamma_{ow} D_p (1 + \cos \theta_{ow}) \sim 10^{-8}$ N, where θ_{ow} is the contact angle of the oil-water

interface with the particle. Due to similar magnitudes, the hydrodynamic forces on the particles can overcome the surface forces holding the particles on the droplet interface.

The above explanations have been added to the supplementary information of the paper for better clarity.

2. The authors explored several potential applications of the composite LM, such as microreactor, capsule, crystallization and cell culture. But some of these applications are already not well established for liquid marbles, let alone oil-coated composite liquid marbles. For example, in figure 5, it was mentioned that the substrate attachment of capsules can be an issue for many applications. So the authors removed the substrate by wet etching with an ammonium persulfate solution, which is quite troublesome.

Response:

In previous reports, LM and droplets have been used in many different applications. Several applications have failed to mature in droplet and LM-based platforms due to the problems of rapid uncontrolled evaporation. The unique ability of our method is to be able to control the evaporation rate by tuning the thickness and material of the oil layer. This ability enables previously unexplored applications such as single crystal growth. It is worth noting that LM-based crystallization has been reported but is limited to interfacial crystallization and polycrystalline salts.¹³⁻¹⁵ To our knowledge, there has been no demonstration of single crystal growth inside LM or composite LM. Control of evaporation is of utmost importance in single crystal growth as it ensures the optimum growth rate.

Single crystal growth can be carried out inside a droplet (or LM) by submerging it inside an immiscible oil.⁵ However, submerging cannot tune the evaporation rate. Submerging provides

an extremely low evaporation rate; thus, the time taken to form a single crystal is very long.¹⁶ As the tunability of the evaporation rate is absent, submerging is not suitable for tuning crystal properties by changing the crystallization rate.^{17,18} Very slow evaporation also creates problems in the initial screening and optimization process, as the time required to grow crystal is also significantly higher.^{19–21} Additionally, harvesting a crystal from oil is very difficult and requires special protocols and complex accessories.^{22,23} Table 2 compares the single crystal growing methods inside droplets.

Single crystal	Concentration (g/ml)	Bare Droplet/LM/composite LM	Submerged Drop in Oil	LMOI (This work)
Copper sulfate	0.3	No	Yes (> 2 months)	Yes (~ 20 hrs)
Rochelle salt	1.5	No	Yes (> 2 months)	Yes (~32 hrs)
Sodium Nitrate	0.9	No	Yes (~ 13 days)	Yes (~ 9 hrs)

Table 2: Possibility of single crystal growth inside drops with various methods. The value in the bracket suggests the typical time frame for the single crystal growth.

Submerged and hanging droplet techniques have also been used in many biological applications, such as cell spheroid growth and protein crystallization.^{24,25} These techniques require a humidity-controlled chamber. Even with humidity as high as 95%, the droplet evaporates faster than LMOI, and the droplet’s lifetime is limited to a few hours (Figure 10). Thus, for long-term cell culture, special arrangements are required. Such as continuous injection or replacement of liquid to balance the evaporation rate.^{25–29} This arrangement needs complex microfluidic devices, and the evaporation rate varies largely with the device designs.^{25,26,28,29} The difficulty in handling and accessing droplets is another major concern in both hanging and submerged droplet techniques.^{28,30} We have demonstrated these applications

to emphasize that LMOI is useful in places where droplet or LM-based techniques have struggled to become established.

Most of the applications demonstrated here were performed without detachment from the substrates. However, in order to increase the utility of the technique, we have demonstrated a technique to isolate the capsules from the substrate without affecting the inner material. We would like to highlight that the wet-etching-based process is scalable (multiple solid-shell droplets can be deposited and released simultaneously). The above explanations have been added to the supplementary information of the paper for better clarity.

Figure 10: Comparison of LMOI evaporation with bare droplets placed in a highly humid environment.

3. The merit of composite LM in this manuscript is the uniform oil layer as said in Line 155.

But there is no experimental evidence.

Response:

To verify the thickness uniformity of the encapsulation layer, we fabricated solid-shell encapsulation capsules with varying particle sizes and mass loadings. After preparing the capsules, we cut them in half and released the trapped liquid. The encapsulation thickness was

measured at three different points using SEM: the bottom left, top, and bottom right, as shown in Figure 11(A). The thickness values obtained from these measurements (3 independent experiments) are presented in Figure 11(B) for the different particle sizes and Figure 11(C) for the different mass loadings. Our analysis revealed that the thickness values for the different sections of the capsules were fairly uniform, indicating that the encapsulation layer thickness is fairly uniform in the capsules. The uniformity is found to be 7.4 % mean thickness deviation for a single droplet measured over three different points and three independent measurements.

Figure 11: (A) SEM image of the half-cut capsule representing the various sections of the capsules, namely, bottom left, Top and Bottom right. Scale bar = $200\ \mu\text{m}$ for the central image. For others = $20\ \mu\text{m}$. (B) The thickness measurement for the bottom left, top, and bottom right sections of the wax capsule for various particle sizes. (C) The thickness measurement for the bottom left, top, and bottom right sections of the wax capsule for various particle mass loading with $35\ \mu\text{m}$ particle size.

Reviewer #3 Comments:

Rutvik et al reported a wicking-assisted cloaking of liquid droplet to produce uniform solid and liquid shell encapsulations (5-200 μm shell thickness for droplet volume spanning over four orders of magnitude). The tunable liquid encapsulation is shown to reduce the evaporation rate of tiny droplets by up to 200 times with a wide tunability in lifetime (1.5 hrs to 12 days). A few engineering demonstrations have been made to showcase the potentials. The topic is interesting, however, this manuscript didn't meet the merits to be published in Nat Comm.

Response:

The major goal behind the work was to create a tunable and uniform coating over sessile droplets. Our studies revealed the critical design and process aspects for creating a uniform & tunable cloaking layer over a droplet. A unique configuration and methodology are required for the formation of crack-free uniform encapsulation. Simply covering the water droplet with an oil droplet leads to large variations in encapsulation thickness along the droplet height. While most of the oil is concentrated near the base, the encapsulation thickness on the top is often limited to 100's nm.⁵⁻⁸ Hence, a composite structure is required to obtain uniform encapsulation.

Most common approaches fail to attain the required encapsulation uniformity and tunability. Encapsulating in oil prior to the particle coating does not form a uniform cloaking as the oil-thickness variation leads to clumping (agglomeration). Further, the encapsulation cannot be performed by pouring the oil on the top of the LM. Oil flow leads to particle detachment, and crack formation takes place. Hence, stable LMOI cannot be formed (Supplementary Video S1). Another way to coat is to introduce oil from the bottom. However, in this case also, the velocity of the oil interface is high and cracks are generated in the particle layer as seen in Supplementary Video S2.

Successful formation of LMOI requires control over the oil flow velocity. To achieve this, we use an oil-infused surface. The nanostructures in the oil-infused surface induces the drag required to slow down the oil flow and hence the process of cloaking. This approach and configuration allow formation of stable and crack-free LMOI. Further, this technique demonstrates encapsulation with both liquid and solid shells. In our opinion, this is the first demonstration of a technique to form tunable and uniform cloaking over a droplet. Further, we demonstrate significant improvements over conventional techniques in several applications. Hence, we believe this work is suitable for consideration in Nature Communications.

The major concern is the lack of novelty. I guess one of the key science here is the cloaking mechanism of oil layer onto the LM, where the Author tried to explain the understanding in Figure 1 and Figure 2. Apart from the observation and empirical summary, the scientific interpretation seems rather slim. It is obvious to me that the cloaking mechanism can be divided into the contact line of oil layer move 'through' the PTFE particle and 'over' the PTFE particle (from Figure 1H). Such transition of liquid-particle interaction is unique from the corresponding macro-scale interactions due to the substantial role of capillary forces. The interaction of a single micro-particle with a dynamic contact line should be traced to uncover the underlying science. The three phase air-oil-LM contact line will be very helpful to elucidate the interaction mechanism and associated intertwined dynamics, including evolution and backward dragging of the transient air-oil-LM contact line, capillary-inertial launch of micro-particles and its subsequent trapping at the air-oil-LM contact line. Based on the force analysis, the Author can provide a model to predict the interesting particle velocity profile and capillary-driven motion during the interaction. Unfortunately, this part of assessment seems totally missing, lead to an incremental scientific advancement in this work.

Response:

As recommended by the reviewer, we have performed additional studies to elucidate the dynamics of the cloaking oil interface.

We agree with the reviewer that the cloaking process can be conceived as a two-step process. In such a process, a rapid covering of the exposed water-air surface by a thin layer of oil is expected within a few milliseconds.³¹ Subsequently, a thicker oil layer is formed by the flow of the oil through and over the particles. However, in our case, we don't observe a two-step process. All our experimental data show a single timescale for the cloaking process. However, our observations don't exclude a two-step cloaking process.

Figure 12: The particle embedded in a wax layer for (A) PTFE (non-spherical) and (B) Glaco-coated glass beads (Spherical). Scale bar = 10 μm .

We also found that the encapsulation thickness is mainly determined by the particle (or agglomerate) thickness. Hence, we assume that most of the flow is primarily through the particles while very little flow is there over the particles. The SEM of the shell also imparts a similar conclusion, where some part of the particle can be seen through the wax layer (Figure 12). Based on the above two observations, we have modeled the cloaking as a single-step

process with the oil thickness approximately same as the particle (agglomerate) thickness, as shown in Figure 13.

Figure 13: (A) The effective forces during the oil rise. (B) The schematic representation of dynamics of oil rise in LMOI. Schematic representation of different surface energy components of (C) LM and (D) LMOI.

We propose a model of oil rising in the LMOI setup. The driving force for oil rise over the liquid marble is capillary force (F_c). Drag offered by the nanostructures in the oil-infused surface (F_d) and weight (W) of the oil are the resistive forces acting against the oil rise (Figure 13(A)). The drag induced by the LM particles is neglected, as they are significantly larger than the nanostructures on the oil-infused surface. Thus, the equation of motion can be given as

$$F_c - W = F_d \quad (1)$$

For silicone oil of 10 cP viscosity and 35 μm particle size ($D_p \sim 2R_p$), the value of the Bond number ($Bo = \rho g D_p^2 / \gamma_o$) and the Galilei number ($Ga = \rho^2 g D_p^3 / \mu^2$) are 6×10^{-4} and $4 \times$

10^{-3} , respectively. Thus, capillary and drag forces are much stronger than the weight of the oil; thus, Eq. (1) can be modified as $F_c = F_d$.

The effective surface energy of the LM and LMOI can be derived by considering individual contributions from water, particle, oil, and interactions between them. Considering n number of spherical solid particles with radius R_p covering the droplet with radius R . The schematic representation of process dynamics is given in Figure 13(B). As represented in Figure 13(B), the whole system can be divided into three major parts, the base, LMOI part (up to which the oil rise happened; height y), and LM part (where oil encapsulation has not happened yet; height $(h - y)$). During the oil rise, the LMOI settles, and the apparent contact angle with the surface changes. thus, the dynamic contact angle θ_d . If the volume of the LMOI is Ω then the effective radius (R) is given by

$$R = \left[\frac{3\Omega}{\pi(2 - \cos\theta_d + \theta_d)} \right]^{\frac{1}{3}} \quad (2)$$

Thus, the surface area of each individual part, the base (S_{base}), LMOI (S_{LMOI}), LM (S_{LM}) and total surface area (S_{tot}) is given by

$$S_{base} = \pi R^2 \theta_d \quad (3)$$

$$S_{LMOI} = 2\pi R^2(1 - \cos\theta_d) - 2\pi R(h - y) \quad (4)$$

$$S_{LM} = 2\pi R(h - y) \quad (5)$$

$$S_{tot} = \pi R^2 \{2(1 - \cos\theta_d) + \theta_d\} \quad (6)$$

We assume that the number of particles in any given configuration is proportional to its surface area. Thus, the number of particles is given by

$$n_{base} = n \frac{S_{base}}{S_{tot}} \quad (7)$$

$$n_{LMOI} = n \frac{S_{LMOI}}{S_{tot}} \quad (8)$$

$$n_{LM} = n \frac{S_{LM}}{S_{tot}} \quad (9)$$

Similarly, the surface fraction (ϕ_s) of individual parts is given by

$$\phi_{s,LM} = \frac{n\pi R_p^2 \theta_w}{S_{tot}} \quad (10)$$

$$\phi_{s,base} = \phi_{s,LMOI} = \frac{n\pi R_p^2 \theta_{ow}}{S_{tot}} \quad (11)$$

Here, the water interface has a contact angle of θ_w with the particles (Figure 13(C)), while the oil-water interface has a contact angle of θ_{ow} with the particles (Figure 13(D)).

As represented in Figure 13(C), the surface energy contribution from the particles submerged in the water can be calculated as $2\pi R_p^2 n \gamma_p (1 + \cos\theta_w)$, where γ_p is the surface energy of the particles. Similarly, surface energy contribution from the particles exposed to air is given by $2\pi R_p^2 n \gamma_{pw} (1 - \cos\theta_w)$, where γ_{pw} is the surface energy of the particle-water interface. The total surface energy of the LM (E_{LM}) part is defined as

$$E_{LM}(y) = S_{LM} \gamma_w \left(1 - \frac{n\pi R_p^2 \theta_w}{S_{tot}} \right) + 2\pi R_p^2 n_{LM} (\gamma_p (1 - \cos\theta_w) + \gamma_{pw} (1 + \cos\theta_w)) \quad (12)$$

Similarly, in the case of LMOI, the water-air surface energy is replaced by water-oil (γ_{ow}), the particle contact angle with the water-oil interface is θ_{ow} , particle-air surface energy is replaced by particle-oil surface energy (γ_{po}) and the additional component of oil surface energy (γ_o) is added (Figure 13(D)). Thus, the effective total surface energy of the LMOI (E_{LMOI}) and the base part (E_{base}) is given by

$$E_{base}(y) = S_{base}\gamma_{ow} \left(1 - \frac{n\pi R_p^2 \theta_{ow}}{S_{tot}} \right) + 2\pi R_p^2 n_{base} (\gamma_{po}(1 - \cos\theta_{ow}) + \gamma_{pw}(1 + \cos\theta_{ow})) \quad (13)$$

$$E_{LMOI}(y) = S_{LMOI} \gamma_o + S_{LMOI} \gamma_{ow} \left(1 - \frac{n\pi R_p^2 \theta_{ow}}{S_{tot}} \right) + 2\pi R_p^2 n_{LMOI} (\gamma_{po}(1 - \cos\theta_{ow}) + \gamma_{pw}(1 + \cos\theta_{ow})) \quad (14)$$

Thus, the total energy variation with y is given by ($E(y)$)

$$E(y) = E_{base}(y) + E_{LMOI}(y) + E_{LM}(y) \quad (15)$$

The capillary force during the oil rise is given by $F_c = -dE(y)/dy$. In order to find the total energy, the variation of experimentally measured θ_d was plotted with the height variation, and then a parabolic fit was carried out (Figure 14(A)). For 10 μl volume and 35 μm particle size, the resulting $E(y)$ and F_c is represented in Figure 14(B) and Figure 14(C), respectively. The variation of energy is nearly linear (Figure 14(B)); thus, the resulting capillary force is constant (Figure 14(C)).

Since the total energy varies linearly, the energy difference between the initial and final state (Δ_{LMOI}) drives the oil rise. The initial state is $E_i = E(0)$, and the final state is $E_f = E(h)$.

Therefore, the driving capillary force can be given by

$$F_c \sim -\frac{dE}{dy} \sim \frac{\Delta_{LMOI}}{h} \sim \frac{E_i - E_f}{h} \quad (16)$$

where h is the height of the droplet, y is the direction of oil rise and Δ_{LMOI} is the surface energy difference between two states.

Figure 14: (A) The parabolic approximation of θ_d with the height variation. (B) The total energy and (C) Capillary force variation with the height of LMOI. The overall variation in the capillary force with height is less than 4 %. The values of the different parameters used while calculating the above plots are $\Omega = 10 \mu\text{l}$, $h = 2.5 \text{ mm}$, $\gamma_p = 20 \text{ mN} \cdot \text{m}^{-1}$, $\gamma_o = 17.4 \text{ mN} \cdot \text{m}^{-1}$, $\gamma_w = 72 \text{ mN} \cdot \text{m}^{-1}$, $\gamma_{ow} = 41.97 \text{ mN} \cdot \text{m}^{-1}$, $\gamma_{pw} = 56 \text{ mN} \cdot \text{m}^{-1}$, $\gamma_{po} = 14.56 \text{ mN} \cdot \text{m}^{-1}$, $\theta_w = 120^\circ$ and $\theta_{wo} = 165^\circ$.

The viscous drag force can be approximated using Poiseuille's law. The velocity gradient in the oil-infused surface is in the order of $\eta V/p$ where η , V , and p are the viscosity of oil, the velocity of oil, and the pitch between the nanostructures. The dissipation occurs over the area $\sim Dy$; thus, the viscous drag force is given by

$$F_d \sim \frac{\eta V D y}{p} \quad (17)$$

Figure 15: (A) The snapshots of oil rise in $10 \mu\text{l}$ LM with $35 \mu\text{m}$ PTFE particles and 50 cP silicone oil. Scale = 1 mm. (B) The evolution of normalized rising height (y/h) with time for different oil viscosity. (C) The scaled graph collapses into a single curve. The solid line has a slope of 0.5. It is provided as a guide.

Balancing the capillary force with the viscous drag force results in

$$V \sim \frac{p\Delta_{LMOI}}{\eta Dh} \quad (18)$$

The velocity of the oil rise is given by $V = y/t$ where t is the time instance of rise. Thus, the rise height (y) is given by

$$y \sim \left(\frac{p\Delta_{LMOI}}{\eta Dh} t \right)^{0.5} \quad (19)$$

Figure 15(A) & (B) represents the rising height of oil over an LM with time. The clear difference is present according to the viscosity of the oil utilized in making LMOI. Nondimensionalization of Eq. (19) reveals the time scale responsible for the oil rise in the

LMOI system (τ_{LMOI}). Thus, normalizing the rise height (y) with the LMOI height (h) gives Eq. (20)

$$\frac{y}{h} \sim \left(\frac{t}{\tau_{LMOI}} \right)^{0.5} \quad (20)$$

where, $\tau_{LMOI} = \frac{\eta D h^3}{p \Delta_{LMOI}}$

As shown in Figure 15(C), all data collapses into a single line when normalized rising height (y/h) is plotted against normalized time (t/τ_{LMOI}). Additionally, the slope of the data is approximately 0.5, which further validates Eq. (20).

With regards to the particle velocity, the careful observation of the particles at the air-oil-LM contact line suggests there is little to no change in particle position during the oil rise. This is due to the jamming of the interface with particles. The SEM image of the hydrophobic glass-bead-based capsule near the air-surface-LMOI contact line reveals a crowding of particles everywhere which restricts any particle velocity present during the oil rise (Figure 16).

The above explanations have been added to the supplementary information and the main manuscript for better clarity.

Figure 16: SEM image of the capsule at the three-phase contact line where crowding of the particle is visible. Scale bar = 100 μm .

In the demonstrations, the authors did not describe what we have learned from this work about general material design/system setting and how we can use these.

Response:

The major goal behind the work was to create a tunable and uniform coating over sessile droplets. Our subsequent experiments on creating a uniform cloaking layer over the droplet revealed the critical design and methodological aspects of the system. Our configuration and methodology are unique, as without them, the stable formation of LMOI is not guaranteed. Coating the oil over a droplet prior to the particle coating does not form a uniform cloaking as the thickness of oil varies with the height of the droplet. This process is also theoretically limited. The coating cannot be performed by directly pouring the oil from the top of the LM as a sudden coalescence shock and particle detachment destabilize the LM, and LMOI cannot be formed (Supplementary Video S1). Another way to coat is from the bottom. However, the velocity of cloaking is very high in such cases, which will generate cracks in the particle layer (Supplementary Video S2). Thus, the successful formation of LMOI requires oil-infused surfaces, which offer additional drag in the spreading of the oil-cloaking layer. This slows down the process of cloaking. The slower spreading of oil ensures stable LMOI formation without any cracks.

Tunable and uniform cloaking of oil over a wide range allowed us to tune the droplet evaporation rate in addition to extending the lifetime. The uniformity of the coating ensures very little to no directional evaporation, thus, extremely useful in controlling the single crystal and cell growth. The tunability aspect ensures precise control of the evaporation rate, which is essential in controlling the properties of crystals. Additionally, tunability and uniformity ensure control over the strength and dissolution of the solid capsule for on-demand delivery.

Even a series of demonstrations have been made, such as crystal growth, biological reactor, and functional platform, limited attractive/innovative points are presented, not mention that these demonstrations are well developed in other liquid droplet related research. May be Author can start to think from some related questions as following, why did the authors choose the engineering concepts? How these concepts compared to the conventional technologies? How about the robustness of each application demo? ...The chemical – physical assessments on the showcased demo seem incomplete, which need a substantial effort to work on it in a systematic way.

Response:

In many previous reports, LM and droplets have been used in many kinds of applications. However, the application of these techniques is limited in many crucial aspects (as seen in Tables 3, 4, 5, & 6). Table 3 enlists droplet evaporation reported in various literature. The sessile droplet placed over an oil-infused surface shows nearly the same lifetime as the bare droplet. Even with relative humidity as high as 95%, the droplet evaporates in 10 hrs. Additionally, LMOI’s other counterparts, such as liquid marble (LM) and composite liquid marble, also suffer from rapid evaporation. The only technique that outperforms LMOI in reducing evaporation is submerging the droplet inside an oil bath. However, in the submerged case the tunability of evaporation is entirely absent in this technique.

Method	Material Used Droplet/Particles/Oil	Evaporation time	Droplet Volume Range	Temperature/Relative Humidity	Particle coating Thickness/sizes	Remarks	Literature References
Sessile Drop	Polystyrene Colloidal solution/ –	~500 sec	0.1 µl	~ 23°C / 47%	NA	Non-uniform coating.	32

coated with Oil	/Silicon Oil 100cst					No tunability.	
	Water/ – /K16256 Oil	~500 sec	0.087 μ l	~ 25°C / 50%	NA		31
	¹ Water/ – /Silicon Oil 350cst	~84 min	4 μ l	~ 22°C / 38 %	NA		7
	² Water/ – /Silicon Oil 1000cst	160 min	10 μ l	–	NA		6
	Water/silicone oil	10 hrs	10 μ l	~ 25°C / 95 %	NA		Experiments by authors
Liquid Marble (LM)	Water/PTFE/ –	26 – 60 min [!]	5 μ l	~ 26°C / 54 -60 %	20– 100 μ m / 5– 9 μ m	NA	33
	Water/Graphene/–	5 – 50 min [!]	5 μ l	~ 23°C / 5 -87 %	– / 2– 30 μ m	NA	34
	PTFE/Agarose Hydrogel sphere	24 hrs	2 to 50 μ l	~ 37°C / NA	1 μ m	Hydrogel sphere inside LM. No tunability	35
	PTFE	72 hrs	10 to 50 μ l	~ 37°C / NA	1 μ m	Floating LM over the water bath. No tunability	36
Submerged LM	CD@POSS particles	~ 1000 times normal LM	NA	~ 26°C / NA	NA	No tunability No Uniformity	16

Composite LM	PTFE/Mineral oil	~ 3 hrs	10 μ l	~ 23°C / 50 %	35 μ m	No tunability No Uniformity	Experiments by authors
LMOI	PTFE particles with silicone/mineral oil or wax	12 days (10 μ l & 35 μ m shell)	14 nl to 200 μ l	~ 25°C / 50 %	5 to 200 μ m	Tunable, Uniform	This work

*NA- not applicable;¹ – time reported before LM buckling;¹ – Oil coated on the hydrophobic surface;² – Oil coated (base coating ~ 16 μ m) on the structured surface;

Table 3: Comparison of evaporation rate by different techniques.

Table 4 reviews various solid shell-making techniques with the liquid core. However, most of the liquid/solid shell encapsulation techniques are either limited by the thickness of the shell or by the volume of the inner liquid. This is because these techniques cannot form tunable and uniform shells over a wide range. Tunability of solid and liquid shells over a wide range allows controlling processes within the encapsulated materials in a better manner. The unique ability of our method to tune and control the thickness (5 to 200 μ m) of the encapsulation layer over a wide range of encapsulated volumes (4 orders of magnitude) enables previously unexplored applications such as single crystal growth.

Work/Literature	Material Used	Capsule size/Volume Range	Shell thickness	Remarks	Reference
Microfluidics	Aqueous poly(vinyl alcohol)	0.17 nl to 8.1 nl	7 to 50 μ m	Permeable shell	37
	Aqueous poly(vinyl alcohol)	0.61 nl to 18 nl	1 to 67 μ m	-	38
	Aqueous poly(vinyl alcohol)	113 nl	7 to 70 μ m	Hollow shell	39
	Palm oil	38 nl	30 to 73 μ m	Hermetic shell	40

	Aqueous poly(vinyl alcohol)	1.4 nl to 3.6 nl	9.7 to 28.4 μm	Permeable shell	41
	PNIPAM gel	0.52 pl to 0.52 μl	NA		42
Droplet Impact/jetting/needle	Alginate drop in calcium chloride solution	1.43 nl to 1.43 μl	140 μm to 1400 μm	Shell thickness depends on capsule volume	43
	Alginate drop in calcium chloride solution	24.4 μl to 44.6 μl	8 to 70 μm	-	44
	Glycerol – water mixture in silicone oil	6.21 μl to 13.43 μl	50 to 190 μm	-	45
	Paraffin Wax	10 μl	700 μm	Hermetic shell	46
	PLA + Silica	0.52 nl to 65 nl	$\sim 10 \mu\text{m}$	-	47
Stamping	Paraffin Wax	10 μl	500 to 1000 μm	Hermetic shell	46
3D printing	PLGA solution	14nl	$\sim 10 \mu\text{m}$	-	48
LMOI	PTFE particles with silicone/mineral oil or wax	14 nl to 200 μl	5 to 200 μm	Hermetic Shell	This work

Table 4: Comparison of different capsule-making techniques.

Table 5 reviews various crystal-growing techniques. The commonly used bulk evaporation often suffers from the lack of evaporation control. Additionally, the desired size and uniformity of the crystal are also hard to achieve. Crystal growth and evaluating reaction kinetics need a controlled evaporation of the droplets. Control of evaporation is of utmost importance in single crystal growth as it ensures the optimum growth rate. Previously reported methods of crystal growth, such as submerging LM or droplets inside an immiscible phase, cannot tune the evaporation rate.⁵ Such techniques provide an extremely low evaporation rate; thus, the time taken to form a single crystal is very long.¹⁶ The evaporation rate tunability is required in creating crystals with different properties.¹⁸ Very slow initial evaporation in submerged droplets also creates problems in the initial screening and optimization process.¹⁹⁻²¹

Moreover, most of the conventional techniques produce multiple small crystals rather than one big crystal, which needs further processing to ensure a larger crystal formation.⁴⁹ Additionally, harvesting a crystal from oil is very difficult and requires special protocols and complex accessories.^{22,23} It is worth noting that LM-based crystallization has been reported but is limited to interfacial crystallization and polycrystalline salt.^{13–15} To the best of the authors' knowledge, there has been no demonstration of single crystal growth inside LM or composite LM.

Method	Material Used	Volume	Advantages	Disadvantages	Literature References
Bulk Evaporation	ϵ -Hexanitrohexaazaisowurzitane	20 ml	Easy, scalable, and fast	Multiple crystal, Crystal density changes with evaporation rate, Difficult to control evaporation	18
	Lysozyme, Alcohol Dehydrogenase, Bovine Serum Albumin	3000 ml	Scalable, simple handling	Multiple crystals, Smaller in size	50
Submerged Droplet (Vapour Diffusion)	Lysozyme	5 μ l	Reduced evaporation rate, Less contamination	Multiple crystals, Shock nucleation, No tunability of evaporation	51
	Thaumatococcus	\sim 4 μ l	Crystallization time can be changed with oil, Batch process	Initial screening and optimization are time taking	20
	Lysozyme, Ferritin, Apoferritin, Glucose isomerase	20 nl to 2000 nl	Crystallization time can be shortened by changing	Multiple crystals, Require robotic systems, Initial screening,	21

			the droplet volume	and optimization are time taking	
	Alcohol Dehydrogenase	2 μ l	Less contamination, resistant to small changes in temperature	Multiple crystals, Special protocol is needed for harvesting; not useful if the crystal has a solubility in oil	22
Liquid Marble	Sodium Chloride	10 μ l	Promote Interfacial crystallization	Not a single crystal, Polycrystalline Salt	13-15
LMOI	Copper sulfate, Rochelle salt, Sodium Nitrate, Lysozyme	14 nl to 200 μ l	Tunable evaporation, Single crystal formation, No merging, Optimal evaporation rate, easy screening, Tunable size	Not useful if the crystal has a solubility in oil, most suitable for water-based solvents.	This Work

Table 5: Comparison of different crystal growth techniques.

Table 6 reviews various platforms for bioreactors. Submerged and hanging droplet techniques have also been used in many biological applications, such as cell spheroid growth and protein crystallization.^{24,25} These techniques require a humidity-controlled chamber. Even with humidity as high as 95%, the droplet evaporates faster, and the droplet's lifetime is limited to a few hours (Figure 10). Thus, for long-term cell culture, special arrangements are required. Such as continuous injection or replacement of liquid to balance the evaporation rate.²⁵⁻²⁹ This arrangement needs complex microfluidic devices, and the evaporation rate varies largely with the device designs.^{25,26,28,29} The difficulty in handling and accessing droplets is another major

concern in both hanging and submerged droplet techniques.^{28,30} Similarly, the cell culture inside LM also suffers from a high evaporation rate. In order to reduce the evaporation rate, various techniques have been proposed, such as encapsulation of hydrogel³⁵ in the LM and stabilizing LM on the water bath.³⁶ However, such arrangements are limited to certain kinds of cell clusters,³⁵ and evaporation is still prevalent. Microfluidics-based cell cultures also suffer from evaporation and large osmolality shift for mammalian cell cultures.⁵²⁻⁵⁴ The LMOI platform does not require any special cumbersome set up to mediate the evaporation rate. Additionally, LMOI possesses higher mechanical strength, which enhances the stability of bioreactors and prevents it from merging. However, on-demand merging is also demonstrated by using the stimuli-responsive oil. It is also possible to sample the liquid from LMOI without disrupting it.

Method	Material Used	Volume	Advantage	Disadvantage	Literature References
Hanging Droplets	A431.H9 cells	10 – 20 μ l	Uniform cluster, High throughput	Require humidification chamber with media reservoir.	26
	HCT-116 eGFP cells	14 μ l	Controllable, reproducible multi-spheroid culture	Require microfabrication and integrated microfluidics to compensate for evaporation	25
	NA	14 μ l	Uniform cluster, multi-spheroidal culture (interconnected droplets)	Needs high humidity (> 95%) with constant feeding for evaporation control, Microfabrication	29
Liquid Marble	Olfactory ensheathing cells/PTFE particles	2 to 50 μ l	Toroidal cluster formation	Require hydrogel inside LM to reduce evaporation. Evaporation is still prominent.	35
	Olfactory ensheathing	10 to 50 μ l	Uniform clusters	Floating LM over the water	36

	cells/PTFE particles			bath. Careful handling is needed. Evaporation is still prominent.	
Microfluidics	Endothelial cell / PDMS	-	High cell volume to extracellular fluid volume	Large surface-to-volume ratio, promotes evaporation, sensitive to mammalian cells (due to osmolality shift in thin PDMS – even with high humidity)	54,55
	Mammalian cells / PDMS	-	High O ₂ and CO ₂ permeability	Chamber collapse due to evaporation, high osmolality shift, water has to pump through near cell culture to reduce osmolality shift	52–54
LMOI	Ovarian Cancer cell & Yeast	14 nl to 200 µl	No need for Evaporation control, Prolonged incubation	Still need media exchange at several intervals	This work

Table 6: Comparison with different cell growth techniques.

A more detailed comparison with the conventional technique of each application is described in Tables 3, 4, 5, & 6. Overcoming such shortcomings requires an engineering innovation that can address most of the issues faced by all related applications. In addition to improvement in many aspects, we demonstrated the robustness of our LMOI platform by testing it for many different applications. Additionally, the mechanical robustness of both liquid and solid shell LMOI has been demonstrated in the article. The temperature stability of the LMOI platform is found to be higher than conventional droplet and LM-based platforms. The scalability of the technique was proven by automating the LMOI formation process and carrying out the single crystal growth with greater than 95% yield.

The above explanations have been added to the supplementary information of the paper for better clarity.

In conclusion, the state of manuscript is pre-mature, which certainly doesn't meet the merits for publishing in Nat Comm. It might fit to some specified journal i.e. Sci. rep. etc. I would recommend to reject or transfer. However, I suggest authors to revise the manuscript based on above comments prior to submit to other journals.

Response:

We express our sincere gratitude to the reviewer for providing us with valuable feedback, which has contributed significantly to the improvement of our manuscript. However, we respectfully hold a different perspective regarding the reviewer's opinion of our paper. In accordance with the reviewer's suggestion, we have incorporated a comprehensive modeling of the oil cloaking dynamics, thereby enriching the manuscript and presenting a more comprehensive body of work. We firmly believe that our study represents a pioneering demonstration of uniform and tunable cloaking of sessile droplets, encompassing both liquid and solid encapsulations. This aspect of our work has far-reaching implications across various application domains, thereby rendering it suitable for publication in esteemed journals such as Nature Communications.

Reviewer #4 Comments:

Summary: This manuscript describes a novel wicking-assisted method of cloaking droplets, its detailed experimental demonstration along with theoretical analysis. Overall, the manuscript is interesting, however, some key components involved in the microparticles design are not sufficiently described or analyzed. It is not too clear what exactly is the groundbreaking novelty of the paper compared to the prior art. The paper should be reconsidered after a major revision, in which the authors clarify the importance of their findings and address the following critical questions.

Response:

The novelty of this work was to create a tunable and uniform coating over sessile droplets. Shell thickness tunability over a range of 5-200 μm for droplet volume spanning over four orders of magnitude was demonstrated. The technique obtains a high thickness uniformity with a mean thickness deviation of 7.4% over multiple droplets and shell-thickness. This study reveals the critical design and process aspects required for creating a uniform and tunable cloaking layer over a droplet. A unique configuration and methodology are required for the formation of crack-free uniform encapsulation. Simply covering the water droplet with an oil droplet leads to large variations in encapsulation thickness along the droplet height. While most of the oil is concentrated near the base, the encapsulation thickness on the top is often limited to 100's nm.⁵⁻⁸ Hence, a composite structure is required to obtain uniform encapsulation.

Previous approaches fail to attain the required encapsulation uniformity and tunability. Coating the oil before particle-coating does not form a uniform cloaking as the oil-thickness variation leads to particle clumping (agglomeration). Encapsulation cannot be performed by pouring the oil on the top of the LM. Oil flow leads to particle detachment. Cracks emerge and LMOI cannot be formed (Supplementary Video S1). Another way to coat is to introduce oil from the

bottom. However, in this case also, the oil velocity is high. This generates cracks in the particle layer as seen in Supplementary Video S2.

Successful formation of LMOI requires the use of oil-infused surfaces with LM. Drag induced by the nanostructures of the oil-infused surface slows down the oil flow and hence the cloaking. Thus, crack-free cloaking is obtained. Further, this technique demonstrates cloaking with both liquid and solid shells. In our opinion, this is the first demonstration of a technique to form tunable and uniform cloaking over a droplet. Further, we demonstrate significant improvements over conventional techniques in several applications. Hence, we believe that this work is suitable for consideration in Nature Communications.

Major Comments/Questions:

1. A detailed discussion is needed regarding the status of PTFE particles on the surface of a droplet, including the immersed percentage of particles, their dispersity (i.e., are they monodispersed on the droplet surface?), the density (i.e., the distance between nearby particles)? The density of the PTFE particles should be measured either by optical means or statistical methods such as checking the mass change post-particle coating.

Response:

Particle density in LM

The particle density (mass loading) of the liquid marble has been varied by coalescing LM with bare droplets. The final volume is fixed at 10 μl (Manuscript Supplementary Figure S4). The different volumes taken for fabricating LM of various mass-loading are listed in Table 7. Where V_{LM} , V_{w} and ϕ_s represent the volume of liquid marble, the volume of the water drop, and the solid surface fraction of the prepared LM. The solid fraction was determined by optical imaging

using a microscope. The mass loading was determined by averaging the mass of ten liquid marbles.

V_{LM} (μL)	V_w (μL)	ML ($\mu\text{g}/\text{mm}^2$)	ϕ_s
0	10	0	0
1.5	8.5	3.12 ± 0.15	0.64 ± 0.04
2	8	5.79 ± 0.13	0.71 ± 0.05
2.5	7.5	6.1 ± 0.39	0.73 ± 0.06
3	7	9.36 ± 0.36	0.84 ± 0.05
5	5	11.14 ± 0.35	0.89 ± 0.04
7	3	12.03 ± 0.43	0.96 ± 0.03
10	0	14.71 ± 0.40	1

Table 7: The values of volume to be taken for particular mass loading and respective solid fraction. The particle used here is $35 \mu\text{m}$ average size (obtained through sieving particles with $25 \mu\text{m}$ and $45 \mu\text{m}$ pore size). As described above, various sizes of LM and droplets collision results in different mass loading (ML).

Structure of LMOI

Directly observing the particles embedded in the cloaking layer is challenging. However, we were able to perform scanning electron microscopy (SEM) analysis on wax-coated LMOI (solid capsule) to gain insights into the distribution of polytetrafluoroethylene (PTFE) particles on the surface of the droplet. The SEM images of the cut capsule revealed that most of the PTFE particles were embedded inside the wax layer (Figure 17(A)). In other words, the particles were not visible on the outer surface of the encapsulation. Additionally, particles were not visible to protrude on the inner surface of the capsule wall (Figure 17(B)). This further indicates the almost complete submergence of PTFE particles within the wax. A similar experiment was performed with Glaco-coated glass beads. We found that a very small amount of particles ($< 10\%$ of the size) is exposed to the air, indicating flow is taking place primarily through the particle layer with some flow over the particles. The particle is observed to protrude

at both the inner and outer surface of the capsule (Figure 16 and 17(C)). The different submergence behavior between the PTFE and glass beads is owing to the differences in their shape and wettability. The PTFE particles are mostly flat with irregular shape. In contrast, glass beads are spherical. Water contact angle on PTFE is 120° whereas on Glaco-coated glass surface it is 150° .

Figure 17: (A) Side view of cut wax capsule for 800 nm particle size. Scale bar = 10 μm . (B) The inner side of the wax capsule where no PTFE particle is observed to protrude. Scale bar = 20 μm . (C) Monodispersity of Glass beads on droplet surface for wax capsule LMOI. Scale bar = 40 μm .

Dispersion

For larger particles, the capsule thickness was nearly equivalent to the particle size, indicating the formation of a monolayer over the droplet interface. In contrast, we observed a much higher capsule thickness for smaller particle sizes (~ 800 nm), suggesting aggregation of 800 nm PTFE particles. For 800 nm PTFE particles, the thickness of the coating was estimated to be around 6 μm (Figure 17(A)). Aggregation for 800 nm particles was confirmed experimentally as shown in Figure 20(C) and Figure 20(D). To directly visualize the monolayer formed over the droplet interface, we used superhydrophobic glass beads coated with Glaco mirror coat spray. The wax capsule prepared by such superhydrophobic glass beads shows a monolayer of the particles at the droplet surface as shown in Figure 16 and Figure 17(C).

Particle density in LMOI

The particle density of LMOI will be slightly different from the respective LM. This is because the LM settles down and contact angle (θ_{LMOI}) decreases during the oil rise. The surface area of the LMOI increases and thus interparticle distance also changes. The particle surface fraction reduces with the LMOI formation. The surface fraction of LMOI (ϕ_{LMOI}) can be approximated by $\phi_{LMOI} = \phi_{LM}A_{LM}/A_{LMOI}$ where A_{LM} and A_{LMOI} are the surface area of LM and LMOI, respectively. Here, $A_{LM} = (6\sqrt{\pi}V_{LM})^{\frac{2}{3}}$ where V_{LM} is the volume of liquid marble and A_{LMOI} is given by

$$A_{LMOI} = (6(1 - \cos\theta_{LMOI}) + \theta_{LMOI}) \left(\frac{\pi}{3}\right)^{\frac{1}{3}} \left[\frac{V_{LMOI}}{(1 - \cos\theta_{LMOI})^2(2 + \cos\theta_{LMOI})}\right]^{\frac{2}{3}}$$

The above explanations have been added to the supplementary information of the paper for better clarity.

2. How the above parameters would influence the stability of liquid encapsulation? For example, how does the density of microparticles influence the cloaking thickness and droplet stability?

Response:

The density (i.e., the distance between nearby particles) affects the capsule thickness. Figure 18 shows the thickness of wax capsules with the change in mass loading for 35 μm PTFE particles. With the increase in mass loading, the interparticle distance decreases, and the thickness of the coating increases. We also found that the LMOI was formed up to a minimum mass loading of 3.12 $\mu g/mm^2$. Below this limit, we observed that cracks appeared promptly upon LM touching the oil-infused surface, indicating that the stability of the LMOI is directly

related to the mass loading of the particles. Above this critical mass loading, all LMOI reported in the paper are mechanically stable.

Figure 18: Variation of wax thickness based on mass loading for 35 μm PTFE particles.

Additionally, if the particles are polydisperse, it is expected to have higher stability than monodisperse particles. This can also be inferred from the data of critical pressure required to break the capsule (Manuscript Figure 5(C)). The capsule made by smaller particles (800 nm) provides nearly similar breakage pressure as 35 μm particles. This is due to the agglomeration of the smaller particles.

The above explanations have been added to the supplementary information of the paper for better clarity.

3. Why were PTFE particles chosen for this method? Is the method generally applicable to other types of hydrophobic microparticles? This requires discussion.

Response:

The PTFE particles were chosen because of their availability. However, we tried different types of hydrophobic particles, such as Glass beads coated with Glaco spray, Lycopodium, and Zein. The proposed method works for all mentioned hydrophobic particles (Figure 19).

Figure 19: The LM (left) and LMOI (right) for different hydrophobic particles.

This data has been added to the supplementary information of the paper.

4. Figure 5C: how can one explain the two different slopes for the pressure before breakage? What is their origin? Can these two different particle size regimes have different stabilizing mechanisms against mechanical stress?

Response:

In our study, smaller particles below 1 μm agglomerate. This is also reflected in the thickness data shown in Manuscript Figure 1(H). To investigate this further, we prepared wax capsules using 35 μm particles and examined them using SEM. The resulting image, presented in Figure 20(A), showed a thickness of around 25 μm , which is less than the average particle size, indicating the absence of agglomeration.

Figure 20: (A) Thickness of the wax capsule for 35 μm particle size. Scale bar = 20 μm . (B) The thickness of the wax capsule for 800 nm particle size. Scale bar = 10 μm . (C) Langmuir-Blodgett method, where particles were first stabilized on a water bath and then transferred onto a solid substrate by carefully dipping the substrate into the liquid surface. (D) Agglomeration size for 800 nm particles lifted on a glass slide using the Langmuir-Blodgett method. Scale bar = 2 μm . (Same as Figure 3 in this response. Reproduced here for convenience of the reviewer)

However, when we prepared wax capsules using 800 nm particles, the resulting thickness was approximately 6 μm , strongly indicating the presence of agglomeration (Figure 20 (B)). This finding was further supported when we collected 800 nm particles from the water interface onto a glass substrate. This was done using the Langmuir-Blodgett method (Figure 20(C)). A layer of 800 nm particles was formed on a liquid bath. Subsequently, the particle film was lifted from the bath on a glass substrate. SEM image of the lifted particle film shows agglomeration with an approximate thickness of 6 μm (Figure 20(D)).

Agglomeration for smaller particles explains the two different slopes in critical pressure data when it is plotted against the particle size. Plotting the critical pressure with the actual shell thickness of the capsule provides a linear trend, which is shown in Figure 21.

Figure 21: Required critical pressure to break the capsule as a function of shell thickness.

5. The lifetime tunability of 10 uL droplets is reported between 1.5 hours to 12 days, what is the resolution of the tunability here - can you program it by the hour, by minutes? Please provide some data and discussion.

Response:

There are several ways by which the droplet lifetime can be tuned.

- (1) Changing the oil type (Manuscript Figure 3(A)),
- (2) Changing the viscosity of oil (Manuscript Figure 3(E)),
- (3) Changing the particle size (Supplementary Figure S11(C)),
- (4) Changing the particle mass loading (Manuscript Figure 3(F)), and
- (5) Decloaking on demand.

Table 8 represents the typical tunability time scale for tunability according to this study. The above explanations have been added to the supplementary information of the paper for better clarity.

Method	Tunability	Remarks
Oil type change	Days	Limited by diffusion and solubility of liquid in the oil
Oil viscosity	Minutes to hours	The viscosity of silicone oil is generally available with 10 cP gaps and thus controllable within several minutes to hours
Particle size	Hours to Days	In the present study, particle size is controllable within 25 μm accuracy; thus, hours to days
Mass Loading	Minutes	Mass loading of the LM can be tuned with a resolution of $\sim 2 \mu\text{g}/\text{mm}^2$. Thus, the tunability is in minutes.
Decloaking of particles on-demand	Minutes	The time when decloaking happens decides the lifetime. Different evaporation rates before and after decloaking.

Table 8: Tunability of droplet lifetime according to methods applied.

6. The figures need to be made clearer by reducing the number of subfigures (ones that are not critical to the main point of the paper can be moved to the SI - such as Fig. 4H, for example).

Response:

Figure 4H has been moved to a supplementary file with figure number “Supplementary Figure”

7. Figure 5 uses the word “stimuli-responsive” but heating to melt the encapsulation is typically not considered a response to a stimulus within active/responsive materials community. Please reword this.

Response:

We have changed the word “stimuli-responsive” to “temperature responsive” throughout the manuscript.

8. Figure 3F should be compared to a control experiment to demonstrate that the Rochelle salt property is only achieved by the method used by the authors.

Response:

The Rochelle salt crystal prepared by bare droplet evaporation (control experiment) results in a polycrystalline structure (Figure 22). Thus, it is not possible to probe the sample and carry out the measurement of the ferroelectric properties because of the random orientation and the fragility of the crystals. Submerged droplets can be used to grow single crystals; however, the time for single crystal growth is considerably higher (> 2 months under 50 cP silicone oil).

Figure 22: Rochelle salt polycrystals prepared from bare droplet evaporation. Scale bar = 200 μm .

Minor comments:

1. Figure 3H demonstrates scalability, but it is not the critical point of this work and can be moved to SI.

Response:

Figure 3H has been moved to a supplementary file with figure number “Supplementary Figure”

2. Please correct “cps” to cP in many instances throughout the manuscript

Response:

We thank the reviewer for the suggestion. We have replaced “cps” with “cP” throughout the manuscript.

References:

1. Gandee, H. *et al.* Unique ice dendrite morphology on state-of-the-art oil-impregnated surfaces. *Proc. Natl. Acad. Sci. U. S. A.* **120**, e2214143120 (2023).
2. James, D. W. The thermal diffusivity of ice and water between -40 and + 60° C. *J. Mater. Sci.* **3**, 540–543 (1968).
3. Mansouri, L., Balistrrou, M. & Baudoin, B. One-dimensional time-dependent modeling of conductive heat transfer during the melting of an initially subcooled semi-infinite PCM. (2017).
4. Murali, G., Mayilsamy, K. & Arjunan, T. V. An Experimental Study of PCM-Incorporated Thermosyphon Solar Water Heating System. <http://dx.doi.org/10.1080/15435075.2014.888663> **12**, 978–986 (2015).
5. Bansal, S. & Sen, P. Axisymmetric and Nonaxisymmetric Oscillations of Sessile Compound Droplets in an Open Digital Microfluidic Platform. *Langmuir* **33**, 11047–11058 (2017).
6. Sahoo, S. & Mukherjee, R. Evaporative drying of a water droplet on liquid infused sticky surfaces. *Colloids Surfaces A Physicochem. Eng. Asp.* **657**, 130514 (2023).
7. Sharma, M., Mondal, S. S., Roy, P. K. & Khare, K. Evaporation dynamics of pure and binary mixture drops on dry and lubricant coated slippery surfaces. *J. Colloid Interface Sci.* **569**, 244–253 (2020).
8. Ge, Q. *et al.* Condensation of Satellite Droplets on Lubricant-Cloaked Droplets. *ACS Appl. Mater. Interfaces* **12**, 22246–22255 (2020).
9. Blanken, N., Saleem, M. S., Antonini, C. & Thoraval, M. J. Rebound of self-lubricating compound drops. *Sci. Adv.* **6**, (2020).
10. Blanken, N., Saleem, M. S., Thoraval, M. J. & Antonini, C. Impact of compound drops: a perspective. *Curr. Opin. Colloid Interface Sci.* **51**, 101389 (2021).
11. Liu, M. *et al.* Improvement of wall thickness uniformity of thick-walled polystyrene shells by density matching. *Chem. Eng. J.* **241**, 466–476 (2014).
12. Quéré, D. Inertial capillarity. *Europhys. Lett.* **39**, 533 (1997).
13. Roy, P. K., Shoal, S., Fujii, S. & Bormashenko, E. Interfacial crystallization in the polyhedral liquid marbles. *J. Colloid Interface Sci.* **630**, 685–694 (2023).
14. Bormashenko, E., Roy, P. K., Shoal, S. & Legchenkova, I. Interfacial crystallization within liquid marbles. *Condens. Matter* **5**, 1–11 (2020).
15. Roy, P. K., Legchenkova, I., Shoal, S. & Bormashenko, E. Interfacial Crystallization

- within Janus Saline Marbles. (2021).
16. Zhao, Z. *et al.* Liquid Marbles in Liquid. *Small* **16**, 2002802 (2020).
 17. Hattori, S., Vandendriessche, S., Koeckelberghs, G., Verbiest, T. & Ishii, K. Evaporation rate-based selection of supramolecular chirality. *Chem. Commun.* **53**, 3066–3069 (2017).
 18. Lee, M. H., Kim, J. H., Park, Y. C., Hwang, J. H. & Kim, W. S. Control of crystal density of ϵ -hexanitrohexaazaisowurzitane in evaporation crystallization. *Ind. Eng. Chem. Res.* **46**, 1500–1504 (2007).
 19. Jancarik, J. & Kim, S. H. Sparse matrix sampling. A screening method for crystallization of proteins. *J. Appl. Crystallogr.* **24**, 409–411 (1991).
 20. D'Arcy, A., Elmore, C., Stihle, M. & Johnston, J. E. A novel approach to crystallising proteins under oil. *J. Cryst. Growth* **168**, 175–180 (1996).
 21. Santarsiero, B. D. *et al.* An approach to rapid protein crystallization using nanodroplets. *J. Appl. Crystallogr.* **35**, 278–281 (2002).
 22. Chayen, N. E. The role of oil in macromolecular crystallization. *Structure* **5**, 1269–1274 (1997).
 23. Douglas Instruments. <https://www.douglas.co.uk/>.
 24. Ferreira, J., Sárkány, Z., Castro, F., Rocha, F. & Kuhn, S. Insulin crystallization: The route from hanging-drop vapour diffusion to controlled crystallization in droplet microfluidics. *J. Cryst. Growth* **582**, 126516 (2022).
 25. Frey, O., Misun, P. M., Fluri, D. A., Hengstler, J. G. & Hierlemann, A. Reconfigurable microfluidic hanging drop network for multi-tissue interaction and analysis. *Nat. Commun.* **2014 51** **5**, 1–11 (2014).
 26. Tung, Y. C. *et al.* High-throughput 3D spheroid culture and drug testing using a 384 hanging drop array. *Analyst* **136**, 473–478 (2011).
 27. Maayani, S., Martin, L. L. & Carmon, T. Water-walled microfluidics for high-optical finesse cavities. *Nat. Commun.* **2016 71** **7**, 1–4 (2016).
 28. Millet, L. J. & Gillette, M. U. Over a Century of Neuron Culture: From the Hanging Drop to Microfluidic Devices. *Yale J. Biol. Med.* **85**, 501 (2012).
 29. Misun, P. M., Birchler, A. K., Lang, M., Hierlemann, A. & Frey, O. Fabrication and operation of microfluidic hanging-drop networks. *Methods Mol. Biol.* **1771**, 183–202 (2018).
 30. Hong, J., Kim, Y. K., Won, D. J., Kim, J. & Lee, S. J. Three-dimensional digital microfluidic manipulation of droplets in oil medium. *Sci. Reports* **2015 51** **5**, 1–11 (2015).
 31. Günay, A. A., Sett, S., Ge, Q., Zhang, T. J. & Miljkovic, N. Cloaking Dynamics on Lubricant-Infused Surfaces. *Adv. Mater. Interfaces* **7**, 2000983 (2020).
 32. Gao, A. *et al.* Control of Droplet Evaporation on Oil-Coated Surfaces for the Synthesis of Asymmetric Supraparticles. *Langmuir* **35**, 14042–14048 (2019).
 33. Tosun, A. & Erbil, H. Y. Evaporation rate of PTFE liquid marbles. *Appl. Surf. Sci.* **256**, 1278–1283 (2009).
 34. Dandan, M. & Erbil, H. Y. Evaporation rate of graphite liquid marbles: Comparison with water droplets. *Langmuir* **25**, 8362–8367 (2009).
 35. Vadivelu, R. K., Kamble, H., Munaz, A. & Nguyen, N. T. Liquid Marble as Bioreactor for Engineering Three-Dimensional Toroid Tissues. *Sci. Reports* **2017 71** **7**, 1–14 (2017).
 36. Vadivelu, R. K. *et al.* Generation of three-dimensional multiple spheroid model of olfactory ensheathing cells using floating liquid marbles. *Sci. Reports* **2015 51** **5**, 1–12 (2015).
 37. Chen, P. W., Erb, R. M. & Studart, A. R. Designer polymer-based microcapsules made

- using microfluidics. *Langmuir* **28**, 144–152 (2012).
38. Xu, S. & Nisisako, T. Polymer Capsules with Tunable Shell Thickness Synthesized via Janus-to-core shell Transition of Biphasic Droplets Produced in a Microfluidic Flow-Focusing Device. *Sci. Reports 2020 101* **10**, 1–10 (2020).
 39. Chen, R., Dong, P.-F., Xu, J.-H., Wang, Y.-D. & Luo, G.-S. Controllable microfluidic production of gas-in-oil-in-water emulsions for hollow microspheres with thin polymer shells. *Lab Chip* **12**, 3858–3860 (2012).
 40. Ryu, S. A. *et al.* Biocompatible Wax-Based Microcapsules with Hermetic Sealing for Thermally Triggered Release of Actives. *ACS Appl. Mater. Interfaces* **13**, 36380–36387 (2021).
 41. Chen, P. W., Brignoli, J. & Studart, A. R. Mechanics of thick-shell microcapsules made by microfluidics. *Polymer (Guildf)*. **55**, 6837–6843 (2014).
 42. Shah, R. K., Kim, J. W., Agresti, J. J., Weitz, D. A. & Chu, L. Y. Fabrication of monodisperse thermosensitive microgels and gel capsules in microfluidic devices. *Soft Matter* **4**, 2303–2309 (2008).
 43. Martins, E., Poncelet, D., Marquis, M., Davy, J. & Renard, D. Monodisperse core-shell alginate (micro)-capsules with oil core generated from droplets millifluidic. *Food Hydrocoll.* **63**, 447–456 (2017).
 44. Bremond, N., Santanach-Carreras, E., Chu, L. Y. & Bibette, J. Formation of liquid-core capsules having a thin hydrogel membrane : liquid pearls. *Soft Matter* **6**, 2484–2488 (2010).
 45. Yin, S. *et al.* Triple-layered encapsulation through direct droplet impact. *J. Colloid Interface Sci.* **615**, 887–896 (2022).
 46. Goertz, J. P., Demella, K. C., Thompson, B. R., White, I. M. & Raghavan, S. R. Responsive capsules that enable hermetic encapsulation of contents and their thermally triggered burst-release. *Mater. Horizons* **6**, 1238–1243 (2019).
 47. Jiang, J. *et al.* High-Throughput Fabrication of Size-Controlled Pickering Emulsions, Colloidosomes, and Air-Coated Particles via Clog-Free Jetting of Suspensions. *Adv. Mater.* **35**, 2208894 (2023).
 48. Gupta, M. K. *et al.* 3D Printed Programmable Release Capsules. *Nano Lett.* **15**, 5321–5329 (2015).
 49. Barros Groß, M. & Kind, M. Comparative Study on Seeded and Unseeded Bulk Evaporative Batch Crystallization of Tetragonal Lysozyme. *Cryst. Growth Des.* **17**, 3491–3501 (2017).
 50. Barros Groß, M. & Kind, M. From microscale phase screening to bulk evaporative crystallization of proteins. *J. Cryst. Growth* **498**, 160–169 (2018).
 51. Blow, D. M., Chayen, N. E., Lloyd, L. F. & Saridakis, E. Control of nucleation of protein crystals. *Protein Sci.* **3**, 1638–1643 (1994).
 52. Forry, S. P. & Locascio, L. E. On-chip CO₂ control for microfluidic cell culture. *Lab Chip* **11**, 4041–4046 (2011).
 53. Thomas, P. C., Raghavan, S. R. & Forry, S. P. Regulating oxygen levels in a microfluidic device. *Anal. Chem.* **83**, 8821–8824 (2011).
 54. Halldorsson, S., Lucumi, E., Gómez-Sjöberg, R. & Fleming, R. M. T. Advantages and challenges of microfluidic cell culture in polydimethylsiloxane devices. *Biosens. Bioelectron.* **63**, 218–231 (2015).
 55. Yun, S. H. *et al.* Characterization and resolution of evaporation-mediated osmolality shifts that constrain microfluidic cell culture in poly(dimethylsiloxane) devices. *Anal. Chem.* **79**, 1126–1134 (2007).

REVIEWER COMMENTS

Reviewer #1 (Remarks to the Author):

The authors have addressed all my concerns and comments and the manuscript has been improved. I have no objection for the publication of this work.

Reviewer #2 (Remarks to the Author):

After reading all the revision files carefully, my overall impression of the revised manuscript is positive. The manuscript offers in-depth analysis and comprehensive characterization of the effects of oil coating on the stability of liquid marble. The study is insightful and solid, and it expands the existing knowledge base of liquid marbles, without introducing groundbreaking new concepts, and contributes to the further exploration of this area. As such, it could be beneficial for scholars and researchers interested in the field.

Reviewer #3 (Remarks to the Author):

The author has made some improvement on the manuscript; however, some queries remain.

The concern on the lack of novelty has not been fully cleared. The key science here is the cloaking mechanism of oil layer onto the LM, it would make sense if author start to compare the cloaking on the plain surface, then the surface with different curvature. The current explanation only apply on droplet surface? Does it work on the plain surface? As previously mentioned, most of the conclusion are supported by the observation and empirical summary, the scientific interpretation seems rather slim.

From the current response, author didn't clarify if the cloaking mechanism is that the contact line of oil layer move 'through' the PTFE particle or 'over' the PTFE particle (from Figure 1H). Maybe it is a mix of both? The current explanation didn't provide a direct evidence, the interaction of a single micro-particle with a dynamic contact line should be traced in situ to uncover the underlying science. I would suggest author to experimentally validate the phenomenon.

While I admit that some classic literature can shine longer, the reference list in this manuscript is a little outdated, most of which were published more than 10 year ago. Some references might be good to enrich the scope:

1. Adv. Funct. Mater. 26, 7206–7223 (2016).
2. Phys. Rev. E 98, 032802 (2018).
3. Nat. Phys. 14, 191–196 (2018).
4. Soft Matter 12, 7632–7643 (2016).
5. Surface Topography: Metrology and Properties 5, 034001 (2017)
6. Small 16 (37), DOI:10.1002/smll.202002802 (2020)

In conclusion, I would recommend a major revision or a transfer.

Reviewer #4 (Remarks to the Author):

The authors thoroughly addressed all the comments in the revised manuscript.

Response to reviewers' comments

Manuscript ID: NCOMMS-23-07654A-Z

“Tunable Cloaking of Sessile Droplets”

We thank the Editor and reviewers for carefully reading our manuscript and providing their insightful comments. We believe these comments will improve the overall quality of the manuscript. Here, we provide a point-to-point response to these comments. For clarity, the reviewers' comments are in blue, and the changes we have made are highlighted in yellow in the revised manuscript.

Reviewer #1 Comments:

The authors have addressed all my concerns and comments and the manuscript has been improved. I have no objection for the publication of this work.

Response:

Thank you for your valuable review and positive feedback on our manuscript. Your input has greatly improved the work.

Reviewer #2 Comments:

After reading all the revision files carefully, my overall impression of the revised manuscript is positive. The manuscript offers in-depth analysis and comprehensive characterization of the effects of oil coating on the stability of liquid marble. The study is insightful and solid, and it expands the existing knowledge base of liquid marbles, without introducing groundbreaking new concepts, and contributes to the further exploration of this area. As such, it could be beneficial for scholars and researchers interested in the field.

Response:

Thank you for your thorough review of the revised manuscript. We are delighted to hear that you have a positive overall impression of our work. We value your insightful comments and recognition of the study's contribution to the existing knowledge base in this area.

Reviewer #3 Comments:

The author has made some improvement on the manuscript; however, some queries remain. The concern on the lack of novelty has not been fully cleared. The key science here is the cloaking mechanism of oil layer onto the LM, it would make sense if author start to compare the cloaking on the plain surface, then the surface with different curvature. The current explanation only apply on droplet surface? Does it work on the plain surface? As previously mentioned, most of the conclusion are supported by the observation and empirical summary, the scientific interpretation seems rather slim.

Response:

We express our gratitude to the reviewer for their valuable feedback. The capillary force governing the oil-cloaking phenomenon relies on the energy difference between the uncloaked and cloaked states. When the oil-infused nanostructured surface contains sufficient oil to cover the entire water interface, these energy states remain unchanged, provided other parameters such as particle/liquid properties and surface area remain unchanged. Consequently, the cloaking process should occur irrespective of the curvature of the interface.

Figure R1: The temporal evolution of the wax infusion in a flat particle-coated water bath. Scale bar = 1 cm.

To ascertain the general applicability of our cloaking method, we conducted oil and wax infusion experiments on a flat interface. Initially, particles were stabilized on a water bath, and a wax-infused surface was then brought into contact with the particle bed. Upon heating the surface, the wax melted, and the infusion occurred inside the porous particle bed with the flat interface as well (Figure R1 and Supplementary Video S8). Furthermore, we measured the average infusion velocity for the flat interface, which was found to be approximately ~ 0.041 mm/s. This velocity is of the same order of magnitude as that observed in the case of low-viscosity oil (10 cP) using the LMOI setup (~ 0.018 mm/s). The slight differences in infusion velocity are attributed to the distinct geometries and the differences in the relative surface areas of the interfaces involved in the experiments. The investigation into the generality of the cloaking method through oil infusion experiments on a flat interface provides compelling evidence that this approach is robust and adaptable across different interfaces, reinforcing its potential for practical applications.

The above information has been added to the **Supplementary Section 14** for clarity.

From the current response, author didn't clarify if the cloaking mechanism is that the contact line of oil layer move 'through' the PTFE particle or 'over' the PTFE particle (from Figure 1H). Maybe it is a mix of both? The current explanation didn't provide a direct evidence, the interaction of a single micro-particle with a dynamic contact line should be traced in situ to uncover the underlying science. I would suggest author to experimentally validate the phenomenon.

Response:

The oil/wax moves primarily through the particles. To investigate our hypothesis, we conducted an in-situ optical imaging analysis of the oil-cloaking process. Figure R2 (and Supplementary Video S3) illustrates the cloaking process of 50 cP oil in Glaco-coated glass

beads, which were stabilized on a flat water interface. The images clearly demonstrate that the oil layer predominantly passes through the gaps between the particles. However, it is worth noting that some flow over the particles can also occur, contingent upon the wettability of the particles.

Figure R2: The cloaking of 50 cP oil in a Glaco-coated glass particle-stabilized on a flat water interface. The dotted line represents the oil cloaked region. Scale bar = 100 μm .

Additionally, as the oil layer passes through the particles, the particle settlement happens. As represented in Figure R3, we tracked the movement of a single particle during the cloaking process. As the oil touches the particle, the particle changes its position due to sudden capillary force by the oil. Additionally, we observed the movement of particles even after oil completely passed through the particle layer. This after movement of particles may arise due to settlement of the particles at the minimum energy position.

Figure R3: The particle tracking during oil movement. The dotted line represents the oil front and red line represents the total path particle took during the cloaking process. Scale bar = 50 μm .

To further validate the observed cloaking behavior, we performed a wax-based encapsulation of the particles and subsequently used scanning electron microscopy (SEM) to examine the results. The SEM images of the Glaco-coated glass beads provide compelling evidence of the complete encapsulation of the glass beads, except for their tops (Figure R4(A) & (B)), thereby confirming the substantial flow of oil through the particles. The SEM image of the PTFE particles reveals some minor protrusion of the PTFE particles through the wax layer, but we have not observed any significant contrast between the particle and wax layer that can conclusively prove about the nature of flow (Figure R4(C)). However, no flow over the Glass beads top conclusively proves that the dominant mechanism of cloaking is flow through the particles and not over them.

The above information has been added to the Supplementary Section 4 for clarity.

Figure R4: (A) The wax infusion in Glaco-coated glass beads where the top of the glass beads appeared to be free from wax. Scale bar = 20 μm . (B) Zoomed in SEM image of the glass beads clearly shows no wax cover over the glass beads. Scale bar = 1 μm . (C) SEM image of the PTFE particles shows protrusion of PTFE through a wax layer. However, there may be wax flow over the particles. Scale bar = 20 μm .

While I admit that some classic literature can shine longer, the reference list in this manuscript is a little outdated, most of which were published more than 10 year ago. Some references might be good to enrich the scope:

1. Adv. Funct. Mater. 26, 7206–7223 (2016).
2. Phys. Rev. E 98, 032802 (2018).
3. Nat. Phys. 14, 191–196 (2018).
4. Soft Matter 12, 7632–7643 (2016).
5. Surface Topography: Metrology and Properties 5, 034001 (2017)
6. Small 16 (37), DOI:10.1002/sml.202002802 (2020)

In conclusion, I would recommend a major revision or a transfer.

Response:

We have now added above mentioned literature to our manuscript.

Reviewer #4 Comments:

The authors thoroughly addressed all the comments in the revised manuscript.

Response:

Thank you for acknowledging our efforts in addressing all your comments in the revised manuscript. We greatly appreciate your valuable feedback, which has helped us enhance the quality of our work.

REVIEWERS' COMMENTS

Reviewer #3 (Remarks to the Author):

This revision has addressed my most comments. With regarding to my reservations on the 'the contact line of oil layer move 'through' the PTFE particle or 'over' the PTFE particle', the current data and response defend the author's point. I am happy with the overall structure and claims.

This is a solid work with current state. I believe that the article can attract and benefit the readers in the fields of surface/interface, droplet/fluids, materials science, and chemical engineering. I would recommend to accept for publishing in Nat Comm.

Response to reviewers' comments

Manuscript ID: NCOMMS-23-07654B

“Tunable Encapsulation of Sessile Droplets”

We thank the Editor and reviewers for carefully reading our manuscript and providing their insightful comments. We believe these comments will improve the overall quality of the manuscript. Here, we provide a point-to-point response to these comments. For clarity, the reviewers' comments are in blue, and the changes we have made are highlighted in yellow in the revised manuscript.

Reviewer #3 Comments:

This revision has addressed my most comments. With regarding to my reservations on the 'the contact line of oil layer move 'through' the PTFE particle or 'over' the PTFE particle', the current data and response defend the author's point. I am happy with the overall structure and claims.

This is a solid work with current state. I believe that the article can attract and benefit the readers in the fields of surface/interface, droplet/fluids, materials science, and chemical engineering. I would recommend to accept for publishing in Nat Comm.

Response:

Thank you for acknowledging our efforts and accepting the article for publication. We greatly appreciate your valuable feedback, which has helped us enhance the quality of our work.